# Histone acetylation orchestrates wound-induced transcriptional activation and cellular reprogramming in Arabidopsis

Bart Rymen [1]*, Ayako Kawamura[1], Alice Lambolez [1,2], Soichi Inagaki [3,4,5], Arika Takebayashi[1], Akira Iwase[1], Yuki Sakamoto[1,2], Kaori Sako[1,6], David S. Favero [1], Momoko Ikeuchi [1], Takamasa Suzuki [7], Motoaki Seki[1,8,9], Tetsuji Kakutani[2,3,4], François Roudier [10] & Keiko Sugimoto [1,2]*

Plant somatic cells reprogram and regenerate new tissues or organs when they are severely damaged. These physiological processes are associated with dynamic transcriptional responses but how chromatin-based regulation contributes to wound-induced gene expression changes and subsequent cellular reprogramming remains unknown. In this study we investigate the temporal dynamics of the histone modifications H3K9/14ac, H3K27ac, H3K4me3, H3K27me3, and H3K36me3, and analyze their correlation with gene expression at early time points after wounding. We show that a majority of the few thousand genes rapidly induced by wounding are marked with H3K9/14ac and H3K27ac before and/or shortly after wounding, and these include key wound-inducible reprogramming genes such as *WIND1*, *ERF113/RAP2.6 L* and *LBD16*. Our data further demonstrate that inhibition of GNAT-MYST-mediated histone acetylation strongly blocks wound-induced transcriptional activation as well as callus formation at wound sites. This study thus uncovered a key epigenetic mechanism that underlies wound-induced cellular reprogramming in plants.

[1] RIKEN Center for Sustainable Resource Science, 1-7-22 Suehiro-cho, Tsurumi, Yokohama, Kanagawa 230-0045, Japan. [2] Department of Biological Sciences, Faculty of Science, The University of Tokyo, 7-3-1 Hongo, Bunkyo-ku, Tokyo 113-8654, Japan. [3] National Institute of Genetics, 1111 Yata, Mishima, Shizuoka 411-8540, Japan. [4] Department of Genetics, School of Life science, The Graduate University for Advanced Studies (SOKENDAI), Shonankokusaimura, Hayama, Kanagawa 240-0193, Japan. [5] PREST, Japan Science and Technology Agency, 4-1-8, Honcho, Kawaguchi, Saitama 332-0012, Japan. [6] Department of Advanced Bioscience, Faculty of Agriculture, Kindai University, Nara 631-8505, Japan. [7] Department of Biological Chemistry, College of Bioscience and Biotechnology, Chubu University, 1200 Matsumoto-cho, Kasugai, Aichi 487-8501, Japan. [8] Plant Epigenome Regulation Laboratory, RIKEN Cluster for Pioneering Research, Wako, Saitama 351-0198, Japan. [9] Kihara Institute for Biological Research, Yokohama City University, Yokohama 244-0813, Japan. [10] Laboratoire Reproduction et Développement des Plantes, ENS de Lyon, UCB Lyon 1, CNRS, INRA, F-69342 Lyon, France. *email: bart.rymen@riken.jp; keiko.sugimoto@riken.jp

Plants regularly experience mechanical wounding and they recover from injuries by regenerating tissues or organs[1–3]. Flowering plants like *Arabidopsis thaliana* (Arabidopsis) start regeneration through either reactivation of relatively undifferentiated cells or dedifferentiation of somatic cells. These cellular reprogramming events often lead to formation of an unorganized cell mass, called callus, from which new organs subsequently develop[4,5]. Under in vitro culture conditions, exogenous plant hormones are used to enhance regenerative capacity, a process referred to as hormone-induced regeneration[4]. We have previously shown that wounding causes dynamic transcriptional changes of genes, ranging from those involved in stress responses to those implicated in metabolic processes or developmental reprogramming[6]. Some of these wound-induced genes function as transcriptional regulators and, among them, genes encoding the ETHYLENE RESPONSE FACTOR (ERF) family proteins, such as DEHYDRATATION RESPONSIVE ELEMENT BINDINGs (DREBs) and RELATED TO AP2.6 (ERF108/RAP2.6), are implicated in general stress adaptation[7]. Other ERF family proteins, such as WOUND INDUCED DEDIFFERENTIATIONs (WINDs), PLETHORAs (PLTs), ERF113/RAP2.6L, and ERF115, are involved more specifically in callus formation and subsequent organ regeneration[6–11]. In addition, wounding triggers expression of genes coding for LATERAL ORGAN BOUNDARIES DOMAIN (LBD) transcription factors, such as *LBD16*, which are involved in hormone-mediated callus formation[6,12].

In the absence of wounding, transcription of reprogramming genes must be tightly regulated to prevent abnormal or inappropriately positioned organ formation, which can occur when these genes are expressed ectopically[13–18]. This regulation is partly mediated by epigenetic mechanisms, which include various types of covalent post-translational modification of histone N-terminal tails[19,20]. Histone modifications modulate the access of the transcriptional machinery to DNA[21–23]. Particularly, trimethylation of the 27th lysine of histone H3 (H3K27me3), catalyzed by POLYCOMB REPRESSIVE COMPLEX 2 (PRC2), maintains a repressive chromatin state for many developmental genes[24,25]. Accordingly, dysfunctional PRC2 results in pleiotropic defects in developmental transitions[26–28]. Strikingly, unscheduled expression of several PRC2 targets causes callus formation from differentiated cells in PRC2 mutants, demonstrating the requirement of PRC2 for cell differentiation maintenance[29,30]. Although genes induced after wounding include several reprogramming genes targeted by PRC2[6], how wound stress relieves their repression is unknown. Previously, transcriptional activation of other PRC2 targets have described a need for loss of H3K27me3 at these loci either through active removal by histone demethylases or passive dilution by rounds of cell division[31–36].

The TRITHORAX GROUP (TrxG) proteins, in contrast, help establishing a permissive chromatin state by catalyzing trimethylation of H3K4 (H3K4me3) and H3K36 (H3K36me3)[37,38]. H3K4me3 is typically enriched near the transcription start site and associated with transcriptional initiation, while H3K36me3 is more broadly deposited and associated with transcriptional elongation[38]. Both H3K4me3 and H3K36me3 are linked to various developmental processes and environmental responses[39] and of note, ARABIDOPSIS TRITHORAX-RELATED 2 (ATXR2)-mediated H3K36me3 is required for hormone-induced callus formation from Arabidopsis leaf explants[40]. Permissive H3K4me3 and repressive H3K27me3 can co-occur, and this mixed chromatin state is thought to control gene expression during seed development and floral transition[41]. The antagonistic relationship between H3K36me3 and H3K27me3 also contributes to the cold-induced transcriptional switch of *FLOWERING LOCUS C* (*FLC*)[42].

Other chromatin factors that promote gene expression include histone acetyltransferases that mediate acetylation of histones H3 and H4. Histone acetylation is thought to neutralize the electrostatic bonds between DNA and histones, leading to chromatin relaxation and thus to a permissive chromatin state[43]. Acetylation of 9th and 14th lysines (H3K9/14) or of 27th lysine (H3K27ac) of histone H3 indeed correlates with transcriptional activation of developmental[44,45] and stress-induced genes[46,47]. In Arabidopsis, four histone acetyltransferases subfamilies exists, GCN5-related N-terminal acetyltransferases (GNATs), MYST (MOZ, Ybf2/Sas3, Sas2 and Tip60)-related, p300/CREB-binding protein (CBP)-related, and transcription initiation factors TAF$_{II}$250-related[48]. Importantly, inhibition of HISTONE ACETYLTRANSFERASE OF THE GNAT/MYST SUPERFAMILY 1/GENERAL CONTROL NONREPRESSED 5 (HAG1/GCN5) or histone deacetylases interferes with hormone-induced callus formation and shoot regeneration in vitro[49,50], suggesting that histone acetylation is essential for transcriptional reprogramming during hormone-induced regeneration.

To uncover the epigenetic mechanisms underlying rapid transcriptional changes occurring upon wounding, we analyzed genome-wide dynamics of gene expression and chromatin landscape associated with permissive H3K4me3, H3K36me3, H3K9/14ac and H3K27ac marks and the repressive H3K27me3 mark. Our chromatin immunoprecipitation (ChIP)-seq and RNA-seq using wounded Arabidopsis roots revealed a positive correlation between wound-responsive gene expression and pre-wound deposition and/or post-wound accumulation of histone acetylation. We further show that inhibition of GNAT-MYST-mediated histone acetylation impedes wound-induced transcription and cellular reprogramming, highlighting a prominent role for histone acetylation in these processes.

## Results

**Wounding induces genes with distinct chromatin states**. We first analyzed the transcriptional changes at 1, 3, 6, and 12 h after wounding in 7-day-old Arabidopsis roots using RNA-seq. The use of roots, instead of hypocotyls as in our previous study[6], allowed us to upscale our sampling capacity to a level sufficient to perform RNA-seq and ChIP-seq in parallel. With this setup, we detected 3665 wound-induced genes and 6010 wound-repressed genes at one or multiple time points after wounding (Fig. 1a, b, Supplementary Fig. 1a, Supplementary Data 1). More than 60% of wound-induced and 40% of wound-repressed genes detected in roots were also up- or down-regulated in hypocotyls upon wounding[6] (Supplementary Fig 1b). In particular, key reprogramming regulators such as *ERF115*, *LBD16*, *PLT3*, *RAP2.6L*, and *WIND1* are induced both in roots and hypocotyls, although the induction of *PLT3*, *RAP2.6L*, and *WIND1* is more transient in roots (Supplementary Fig. 1c).

To determine the chromatin status of genes before wounding, hereinafter referred to as pre-wound status, we performed ChIP-seq using intact Arabidopsis roots and measured the enrichment of H3K27me3, H3K36me3, H3K4me3, H3K9/14ac, and H3K27ac relative to histone H3 levels. We detected 5,777 genes marked with H3K27me3, 12,379 genes with H3K36me3, 13,947 genes with H3K4me3, 12,459 genes with H3K9/14ac and 13,703 genes with H3K27ac (Supplementary Fig. 2a, Supplementary Data 2), which largely overlap with previous ChIP-seq datasets obtained in Arabidopsis[51,52] (Supplementary Fig. 2b). We first examined the chromatin status of 3665 wound-induced genes and found that over 76% of them are marked with at least one of the histone modifications we analyzed (Fig. 1a). As expected, we observed a negative correlation between H3K27me3 levels and gene expression before wounding, while we found a positive correlation

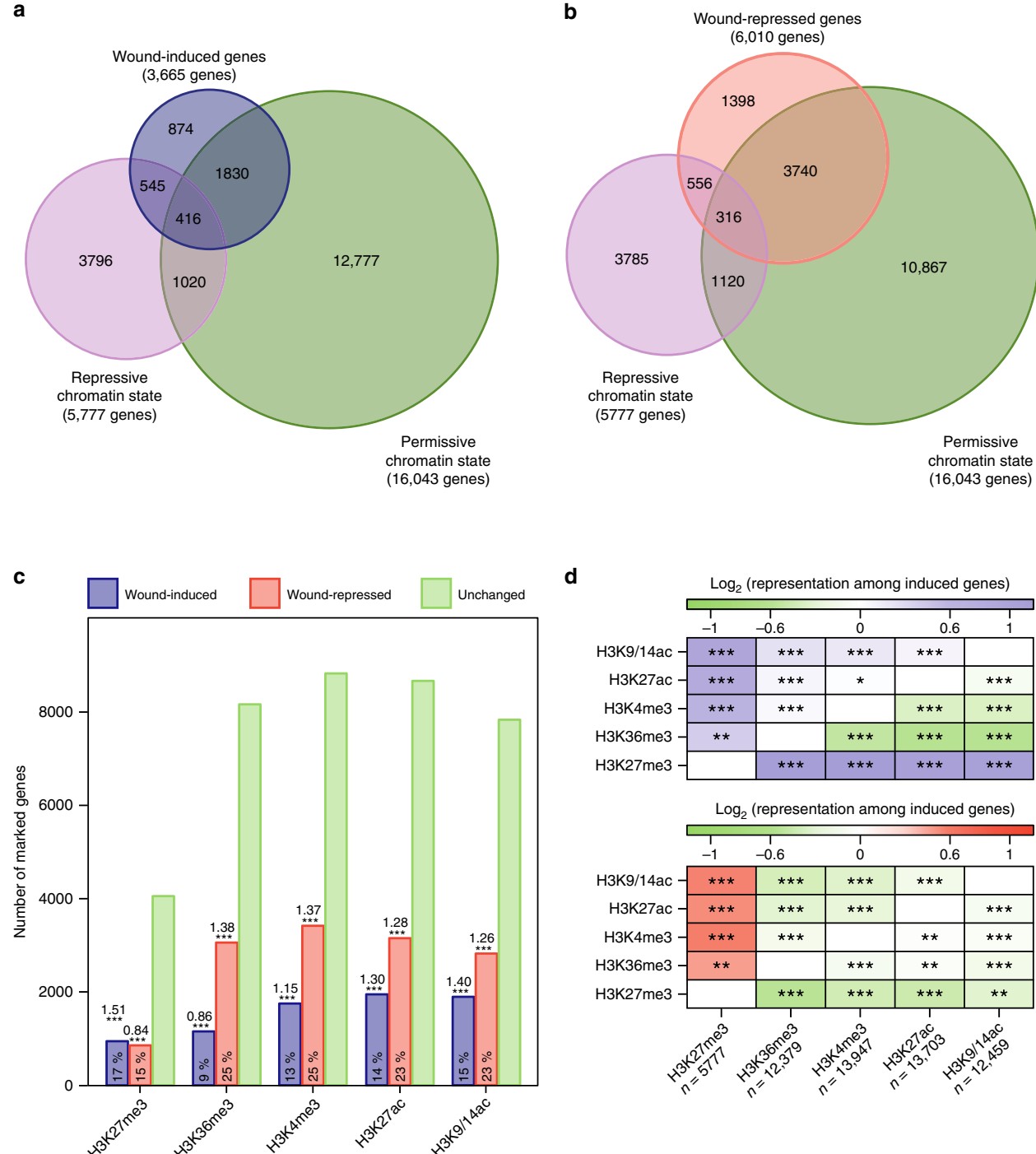

**Fig. 1** Chromatin status of genes before wounding and their transcriptional responses to wounding. **a**, **b** Venn diagram representation of the pre-wound permissive (H3K36me3, H3K4me3, H3K9/14ac, or H3K27ac) or repressive (H3K27me3) chromatin states for **a** 3665 wound-induced genes and **b** 6010 wound-repressed genes. Wound-induced genes include those that show transcriptional up-regulation with a fold change (FC) > 1.5 and a false discovery rate (FDR corrected $p$ values, edgeR test) < 0.001 at one or multiple time points after wounding. Wound-repressed genes include those that show a transcriptional down-regulation with FC < −1.5 and FDR < 0.001 at one or multiple time points after wounding. **c** Distribution of wound-induced, wound-repressed and unchanged genes among genes marked with H3K27me3, H3K36me3, H3K4me3, H3K27ac, or H3K9/14ac. Percentages indicate the ratio of genes induced or repressed by wounding among all genes containing the given mark. Bold numbers indicate representation values, i.e. the ratio between the representation of marked genes among wound-induced or -repressed genes and their representation across all genes. *$p < 0.05$, **$p < 0.01$, ***$p < 0.001$ (hypergeometric test). **d** Heat-map representation of the over- and under-representation of co-marked genes within the 2,791 wound-induced genes and 4,612 wound-repressed genes bearing histone marks before wounding, compared to the occurrence of co-marking among all marked genes. Colors indicate the representation value for each co-occurring modification. $n$—the number of marked genes used as a reference for the enrichment analysis. *$p < 0.05$, **$p < 0.01$, ***$p < 0.001$ (hypergeometric test)

between H3K36me3, H3K4me3, H3K9/14ac, and H3K27ac levels and gene expression (Supplementary Fig. 2c). Interestingly, wounding induces 961 out of 5777 genes bearing pre-wound H3K27me3 within 12 h (Fig. 1a, c), suggesting that a significant proportion (17%) of PRC2 targets may be induced upon wounding. Additionally, genes marked with pre-wound H3K4me3, H3K9/14ac, or H3K27ac are also significantly over-represented among wound-induced genes (Fig. 1c). In contrast, genes associated with pre-wound H3K36me3 are significantly underrepresented among wound-induced genes presumably because these genes are already actively transcribed before wounding (Fig. 1c). Consistently, gene ontology (GO) analysis showed that GO terms "response to stress" or "response to abiotic or biotic stimulus" are enriched among genes associated with pre-wound H3K27me3, H3K9/14ac, H3K27ac, or H3K4me3 but underrepresented among those with pre-wound H3K36me3 (Supplementary Fig. 2d).

We next examined whether the position of these histone marks along target loci affects the responsiveness of gene expression. As previously reported[39,51], most of H3K27me3 is found within gene bodies whereas H3K4me3, H3K9/14ac, and H3K27ac deposition overlaps the transcriptional start sites (Supplementary Fig. 3a). Overall positions of these histone marks do not differ between wound-induced and non-induced genes (Supplementary Fig. 3a), indicating that this parameter is not a determinant for wound-induced transcriptional activation. We also tested whether the initial enrichment levels of these histone marks differ between wound-induced and non-induced genes and whether it correlates with differences in inducibility. Although relative enrichment levels in our dataset are averaged among heterogenous cell populations, we found that H3K27me3 level among 961 wound-induced genes tend to be lower than that of 4816 non-induced genes (Supplementary Fig. 3b), implying that genes with high H3K27me3 level tend to remain transcriptionally non-responsive to wounding. In contrast, overall levels of H3K4me3, H3K9/14ac, and H3K27ac are higher for wound-induced genes compared to non-induced genes (Supplementary Fig. 3b), suggesting that genes with high initial levels of these permissive marks are prone to wound-induced activation.

As previously reported[38], our ChIP-seq dataset suggests that many of these histone modifications co-occur (Supplementary Fig. 2a). Interestingly, 25% of H3K27me3 target genes (1436 genes out of 5777 genes) also possess at least one of the active marks and 29% of this subset of genes (416 genes out of 1436 genes) are wound-induced (Fig. 1a). Enrichment analyses indeed revealed that genes co-marked with H3K27me3 and permissive marks are significantly overrepresented among wound-induced genes (Fig. 1d). In contrast, wounding induces only 13% of the genes bearing H3K27me3 alone (545 genes out of 4341 genes) (Fig. 1a). These observations imply that PRC2 target genes are more likely wound-inducible when associated with additional permissive marks.

**Rapid gene induction correlates with pre-wound H3K9/14/27ac**. To further assess how pre-wound deposition of histone marks correlates with transcriptional activation after wounding, we tested whether pre-wound levels of H3K27me3, H3K36me3, H3K4me3, H3K9/14ac, and H3K27ac affect the timing of wound-responsive gene expression. We grouped the 3665 wound-induced genes in our RNA-seq dataset using a K-means clustering approach and identified eight clusters depicting early to late transcriptional induction (Fig. 2a, Supplementary Data 1). GO analysis revealed that genes induced relatively early, i.e. those up-regulated within first 3 h, are enriched for genes acting in response to stress, while genes induced later include those

implicated in various metabolic processes (Supplementary Fig. 4).

To determine which pre-wound histone modification correlates with early transcriptional activation, we ranked the 3665 wound-induced genes according to their pre-wound relative enrichment levels and plotted the expression clusters to which they belong. As shown in Fig. 2b, we found an almost linear relationship between pre-wound H3K9/14ac levels and the timing of expression when smoothed with the LOESS algorithm. We did not observe such a linearity when we ranked the induced genes according to other pre-wound marking levels (Fig. 2b), suggesting that early transcriptional induction correlates best with pre-wound H3 acetylation. Additionally, we performed a principal component analysis and evaluated the statistical variance of the relative enrichment level for each pre-wound histone mark in comparison to the variance of genes' presence in the different expression clusters, used as a proxy for the "timing" of expression, as well as their expression level before wounding (expression (0 h)). As shown in Supplementary Fig. 5a, within PC1 and PC2 that explain 53.6% and 17.2% of variance, respectively, pre-wound H3K27me3 and H3K36me3 marks display the largest difference, thereby confirming their antagonistic behavior. Similarly, the axis representing "expression (0 h)" anti-correlates with pre-wound H3K27me3, validating the negative correlation between these two parameters. Additionally, the axis representing "timing" is anti-parallel to the axes representing pre-wound H3K27ac and H3K9/14ac (Supplementary Fig. 5a), supporting that early transcriptional induction correlates best with pre-wound H3 acetylation. Plotting the average relative level of histone marks within each cluster revealed that genes in clusters 1–3 display high levels of pre-wound H3K27ac and H3K9/14ac, whereas genes in clusters 7 and 8 tend to have high levels of pre-wound H3K27me3 (Supplementary Fig. 5b), suggesting that high H3K27me3 levels delay transcriptional activation. The deviation across clusters is less pronounced for levels of pre-wound H3K4me3 and H3K36me3 (Supplementary Fig. 5b), further suggesting that these marks do not strongly affect the timing of gene expression.

**Gene induction coincides with H3K9/14/27ac and H3K4me3**. To investigate whether wound stress modulates chromatin states and how this variation correlates with wound-induced gene expression, we examined the level of H3K27me3, H3K36me3, H3K4me3, H3K9/14ac, and H3K27ac at 1, 3, and 6 h after wounding. Several studies used an exogenous epigenome as a reference to compare histone modification levels between samples as histone modification levels may differ extensively[53,54]. However, when the modification levels remain comparable between samples, many studies examined H3 modification relative to histone H3 levels to detect locus-specific alterations[55–59]. Given that overall levels of histone marks do not change drastically within our time points based on a western blot analysis (Supplementary Figs. 6, 12), we normalized our data to histone H3, and evaluated changes in their relative enrichment levels at specific loci compared to unwounded plants. We found 7773 genes with modified enrichment level for H3K27me3, H3K36me3, H3K4me3, H3K9/14ac, and/or H3K27ac after wounding (Supplementary Data 3). We then analyzed whether a loss of repressive marks (H3K27me3) or gain of permissive marks (H3K36me3, H3K4me3, H3K9/14ac, and/or H3K27ac) accompanies the transcriptional activation of the 3665 wound-induced genes. As shown in Fig. 3a, we identified 1731 wound-induced genes that show significant changes in the level of at least one of these histone modifications (Supplementary Data 4). Among these genes, we found that those showing increased relative enrichment level of H3K9/14ac, H3K27ac, and H3K4me3 are

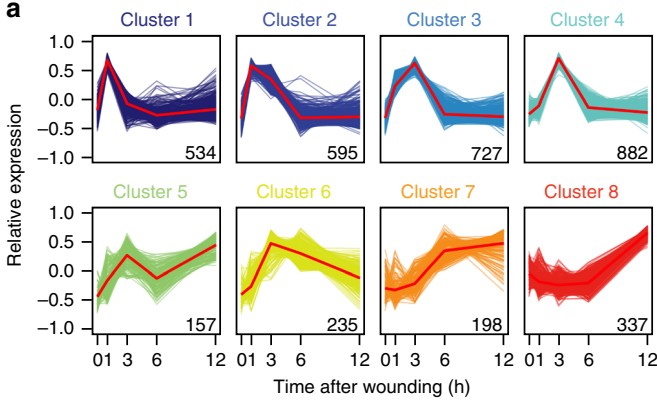

**Fig. 2 Genes rapidly induced by wounding are marked with H3K9/14ac and H3K27ac before wounding. a** K-means clustering of genes significantly induced by wounding (FC > 1.5, FDR corrected *p* values (edgeR test) < 0.001, *n* = 3665 genes). Clusters are ordered from most rapidly (cluster 1) to most slowly (cluster 8) induced genes. The number of genes included in each cluster is indicated and red lines show the average level of their expression. **b** Heat map-based comparison of the relative enrichment levels of pre-wound H3K27me3, H3K36me3, H3K4me3, H3K27ac, and H3K9/14ac and the timing of transcriptional induction after wounding, using a LOESS regression. Genes are sorted based on their relative enrichment levels of pre-wound H3K27me3, H3K4me3, or H3K9/14ac

overrepresented, as 1639 genes (accounting for 45% of 3665 wound-induced genes, representation factor = 4.8 and *p* value ≈ 0) gain either one or a combination of these marks (Fig. 3a). In contrast, we observed that only 3.2% (116 genes, representation factor = 0.8 and *p* value < 1.5 × 10⁻²) of wound-induced genes gain H3K36me3, and that 2.2% (80 genes, representation factor = 1.3 and *p* value < 1.1 × 10⁻²) lose H3K27me3 (Fig. 3a), indicating that these changes poorly contribute to wound-induced transcriptional activation. We observed similar trends

with wound-induced PRC2 target genes (Fig. 3b), thus high-lighting that accumulation of H3K9/14ac, H3K27ac, and H3K4me3, rather than loss of H3K27me3, correlates with their wound-induced transcriptional activation.

To explore the correlations between histone marking and transcriptional changes over time, we plotted the level of wound-induced histone modification for all genes at 1, 3, or 6 h after wounding and visualized the level of their transcriptional response (Fig. 3d). We observed an association between an

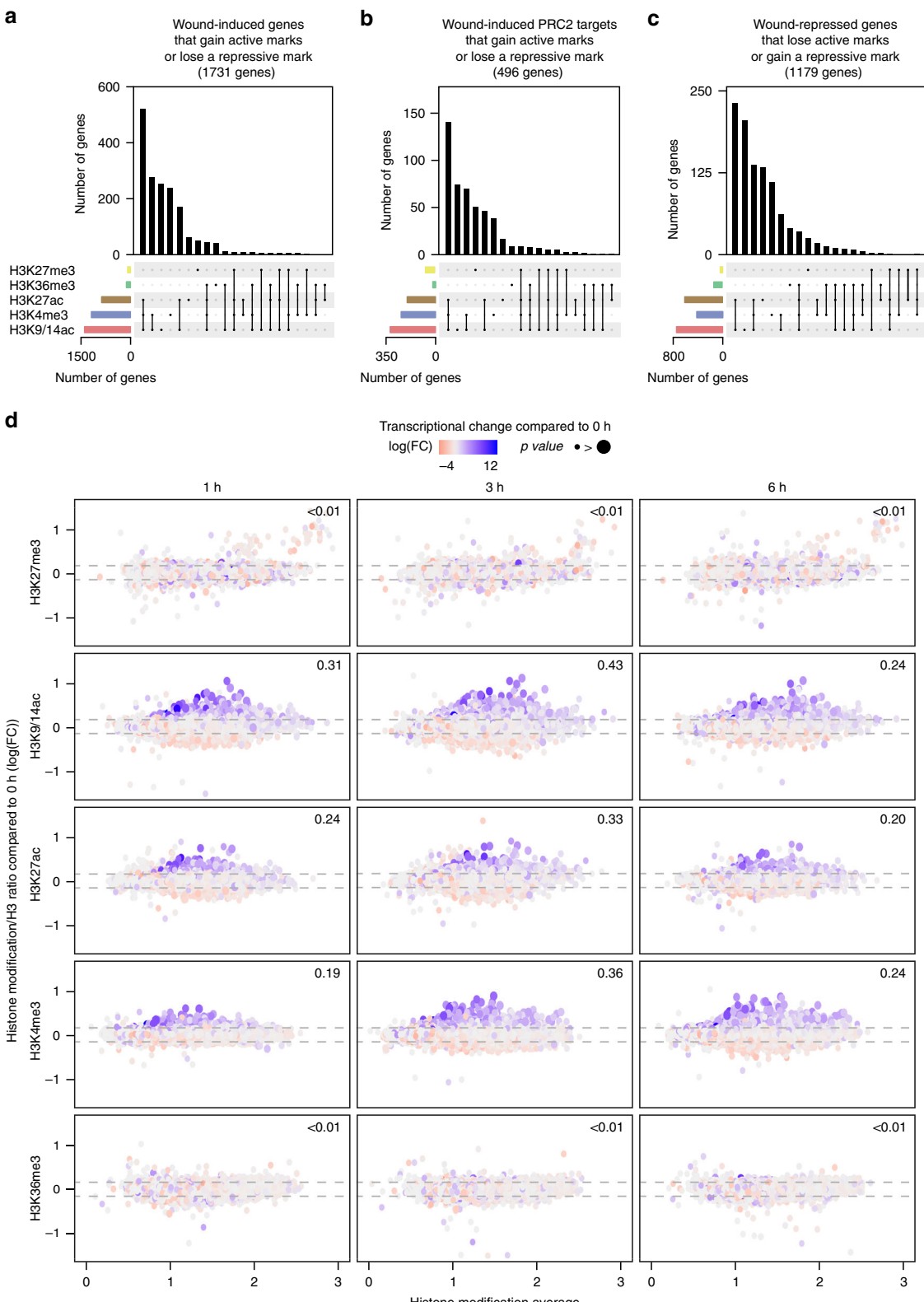

increase in H3K9/14ac and H3K27ac and transcriptional activation as early as 1 h after wounding (Fig. 3d). This coupling becomes stronger at 3 h after wounding, then weakens by 6 h (Fig. 3d). We also observed an association between an increase in H3K4me3 level and transcriptional activation, although this correlation is relatively mild at 1 h and becomes stronger at 3 and 6 h (Fig. 3d). As expected, changes of H3K27me3 and H3K36me3

level are less pronounced at all tested time points, and hardly any correlate with transcriptional changes (Fig. 3d).

## Gene repression coincides with H3K9/14/27ac and H3K4me3 loss. Using these ChIP-seq and RNA-seq data, we investigated how pre-wound and post-wound histone marks correlate with

**Fig. 3** Transcriptional activation after wounding is associated with increase in H3K9/14ac, H3K27ac and H3K4me3. **a–c** Upset plots displaying (**a**, **b**) an increase in H3K36me3, H3K4me3, H3K27ac, or H3K9/14ac or decrease in H3K27me3 for **a** 1731 wound-induced genes or **b** 496 wound-induced PRC2 target genes, and **c** a decrease in H3K36me3, H3K4me3, H3K27ac, or H3K9/14ac or increase in H3K27me3 for 1179 wound-repressed genes. The y-axis shows the number of genes that undergo one or a combination of histone modification, as indicated by the lines below the x-axis. Differentially modified genes correspond to those that show changes with $\log_2(FC) > |0.15|$. **d** MA plots with M (histone modification ratio) and A (histone modification average) displaying the fold changes in H3K27me3, H3K9/14ac, H3K27ac, H3K4me3, and H3K36me3 levels compared to pre-wound histone marking levels, for all genes at 1, 3, and 6 h after wounding. Color and size of dots represent fold change ($\log(FC)$) and significance (FDR corrected $p$ values (edgeR test)), respectively, of the transcriptional response compared to pre-wound transcript levels at the same time point. Numbers in the panel indicate the Pearson coefficient for a linear regression between the fold change in histone modification and the fold change of expression at the indicated time point compared to 0 h

wound-induced transcriptional repression. Among 6010 wound-repressed genes, we found that 4612 genes are marked before wounding with at least one of the histone modifications we analyzed (Fig. 1b, Supplementary Data 2). As expected, genes marked with pre-wound H3K27me3 are underrepresented (Fig. 1c) likely because many of these genes are not expressed before wounding (Supplementary Fig. 2c). In contrast, genes with pre-wound permissive marks are overrepresented among the wound-repressed genes (Fig. 1c). Similar to the wound-induced genes, we detected a significant enrichment of co-marked genes within the wound-repressed genes, among which those marked with H3K27me3 and permissive marks are highly represented (Fig. 1d). To test whether the relative levels of histone modification before wounding correlate with the timing of repression, we grouped 6010 wound-repressed genes into eight clusters defined by their repression dynamics (Supplementary Fig. 7a, Supplementary Data 1). None of the pre-wound levels of the analyzed marks correlated with the expression cluster to which the marked genes belonged (Supplementary Fig. 7b). Moreover, GO enrichment analysis performed on each of these clusters revealed that they represent different sets of biological functions (Supplementary Fig. 8), suggesting that neither the timing of wound-induced repression nor the functional class of wound-repressed genes correlate with pre-wound enrichment levels of chromatin modifications we tested.

Furthermore, we detected post-wound alterations of histone mark levels on only 20% of 6010 wound-repressed genes (Supplementary Data 5). Similar to the wound-induced genes, H3K9/14ac, H3K27ac, and H3K4me3 marks are most responsive to wounding for the repressed genes, as 1106 genes show decrease in at least one of H3K9/14ac, H3K27ac, or H3K4me3 marks (representation factor = 3.3 and $p$ value ≈ 0, Fig. 3c). In contrast, only 143 wound-repressed genes lose H3K36me3 upon wounding (representation factor = 1.6 and $p$ value < $3.5 \times 10^{-9}$), and 37 gain H3K27me3 (representation factor = 0.1 and $p$ value < $2.7 \times 10^{-83}$) (Fig. 3c), confirming that post-wound modification of these two latter marks poorly correlate with transcriptional changes occurring within 12 h after wounding.

**H3K9/14ac precedes or coincides with H3K4me3 and gene induction.** We further explored how pre-wound deposition and post-wound accumulation of H3 acetylation, and post-wound accumulation of H3K4me3, relate to temporal transcriptional dynamics upon wounding, as these are the three parameters that best correlate with wound-induced transcriptional activation. For simplification, we used H3K9/14ac as representative for H3 acetylation, since H3K9/14ac and H3K27ac levels show similar dynamics and changes in H3K9/14ac are more pronounced than those in H3K27ac (Fig. 3d). By examining the combinatory alteration in histone modification for 3665 wound-induced genes, we found that 67% of them already possess H3K9/14ac before wounding and/or accumulate H3K9/14ac after wounding and that, among them, 43% concomitantly gain H3K4me3 (Fig. 4a).

In contrast, only 2% of wound-induced genes gain H3K4me3 without gaining H3K9/14ac (Fig. 4a), supporting the idea that H3K4me3 alone is not a major contributor for wound-induced transcriptional activation. By plotting these histone modification dynamics for wound-induced genes in clusters 1–8, we further uncovered that around 55% of the genes that are most rapidly induced after wounding, i.e. those in cluster 1, already possess H3K9/14ac before wounding, and that the majority of them are transcriptionally activated without gaining H3K4me3 or H3K9/14ac upon wounding (Fig. 4b). In contrast, 40–50% of the genes induced later, in particular those in clusters 2–4, gain H3K9/14ac (with or without pre-wound H3K9/14ac deposition), and this slower activation is better associated with concurrent accumulation of H3K4me3 (Fig. 4b). This notion is supported when grouping genes that gain H3K4me3 (with or without H3K9/14ac accumulation), as these genes are strongly represented in clusters 2–4 (Supplementary Fig. 9). Finally, the largest proportion of genes induced at 6 h or later, i.e. those in clusters 7 and 8, do not experience significant changes in H3K9/14ac or H3K4me3 levels within the first 6 h (Fig. 4b).

To examine the temporal relationship between H3K9/14ac and H3K4me3 dynamics and transcriptional activation, we plotted their average enrichment levels over time and examined how their changes correlate with expression of genes in clusters 1–8 (Fig. 5a). For genes marked with constant levels of H3K9/14ac before and after wounding, we detected only very mild changes in H3K4me3 levels, irrespective of their timing of induction (Fig. 5a). In contrast, for genes that gain H3K9/14ac after wounding, we found a tight correlation between increase in H3K9/14ac and transcriptional activation (Fig. 5a). Importantly, we also observed a delayed increase in H3K4me3 levels that often reach its maximum after the transcriptional peak and this trend was most notable for the rapidly induced genes in clusters 1 and 2 (Fig. 5a). In support of this observation, 212 wound-induced genes gain H3K9/14ac before H3K4me3 accumulation, while only 28 genes gain H3K4me3 before H3K9/14ac (Fig. 5b, c). Altogether, these data show that accumulation of H3K9/14ac often precedes or coincides with H3K4me3 deposition and transcriptional activation.

Furthermore, we tested how these correlations between H3K9/14ac and H3K4me3 dynamics and transcriptional activation apply to a selection of transcription factors involved in cellular reprogramming and/or wound response. As shown in Fig. 5d, genes that are already expressed to some level before wounding and up-regulated by 1 h after wounding, such as WIND1 and LBD16, possess high levels of H3K9/14ac and H3K4me3 before wounding, and these levels remain unchanged. In contrast, genes that are barely expressed before wounding and induced by 3 h, such as RAP2.6, RAP2.6L, and DREB2D, gain H3K9/14ac at 1 h after wounding, followed by accumulation of H3K4me3 after 3 h (Fig. 5d). In addition, these data show that bearing some levels of H3K27me3 has no obvious negative impact on transcriptional responsiveness, as LBD16, RAP2.6L, and DREB2D are all marked

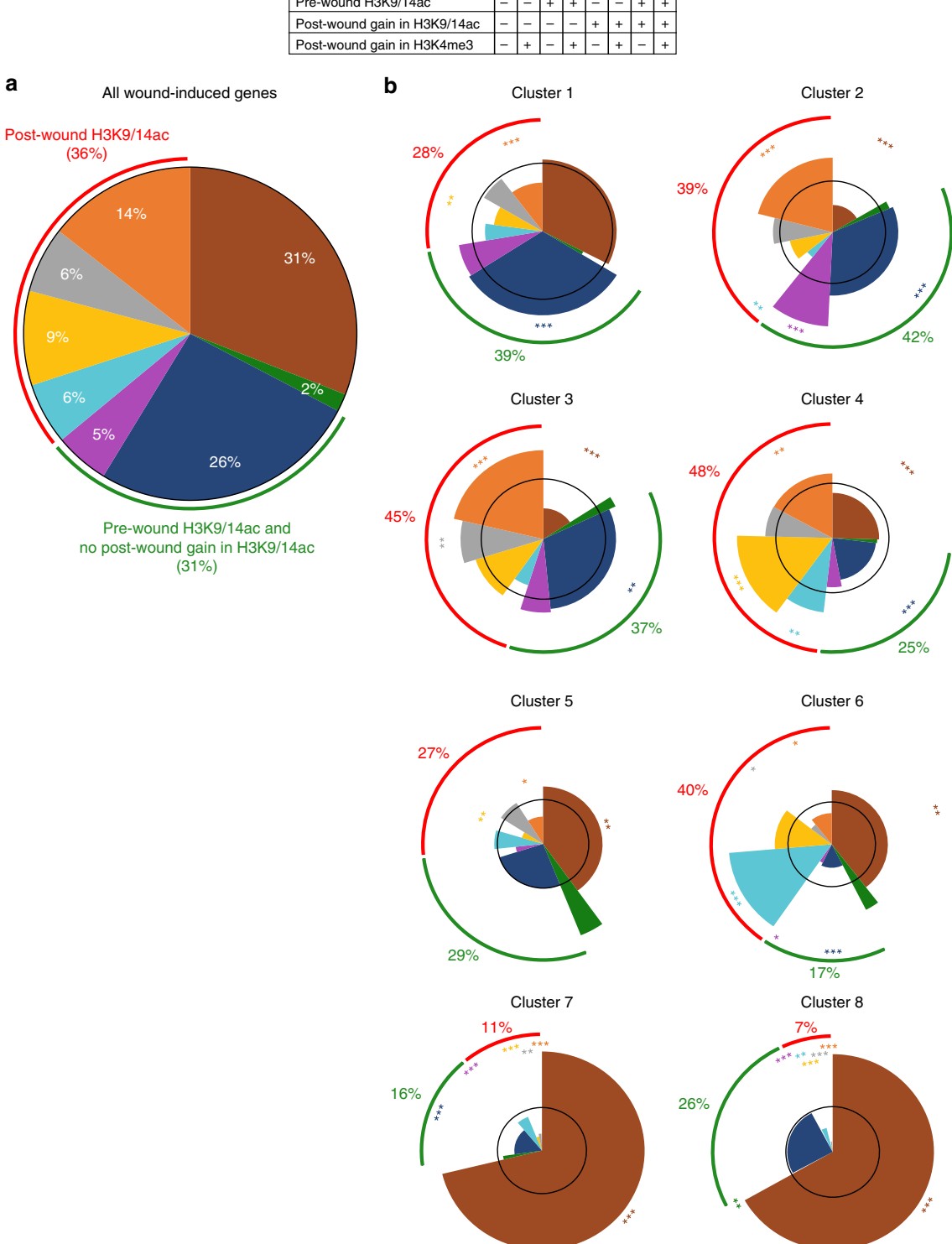

**Fig. 4** Wound-induced increase of H3K9/K14ac marking is associated with slower transcriptional activation. **a** Pie chart representing the percentage of genes associated with pre-wound H3K9/K14ac, post-wound H3K9/14ac and/or H3K4me3 among 3665 wound-induced genes. Genes are grouped based on their association with H3K9/14ac. **b** Spie chart representing the percentage of genes associated with pre-wound H3K9/K14ac, post-wound H3K9/14ac and/or post-wound H3K4me3 among genes within clusters 1–8. Genes are grouped based on their association with H3K9/14ac. The radii of the wedges correspond to the representation factor (hypergeometric test) of the epigenetic category in the cluster compared to its representation among all wound-induced genes. The black circle corresponds to a representation factor of 1 so that wedges inside the circle depict an underrepresented category and wedges that extend beyond the circle depict an overrepresented category. *$p < 0.05$, **$p < 0.01$, ***$p < 0.001$ (hypergeometric test)

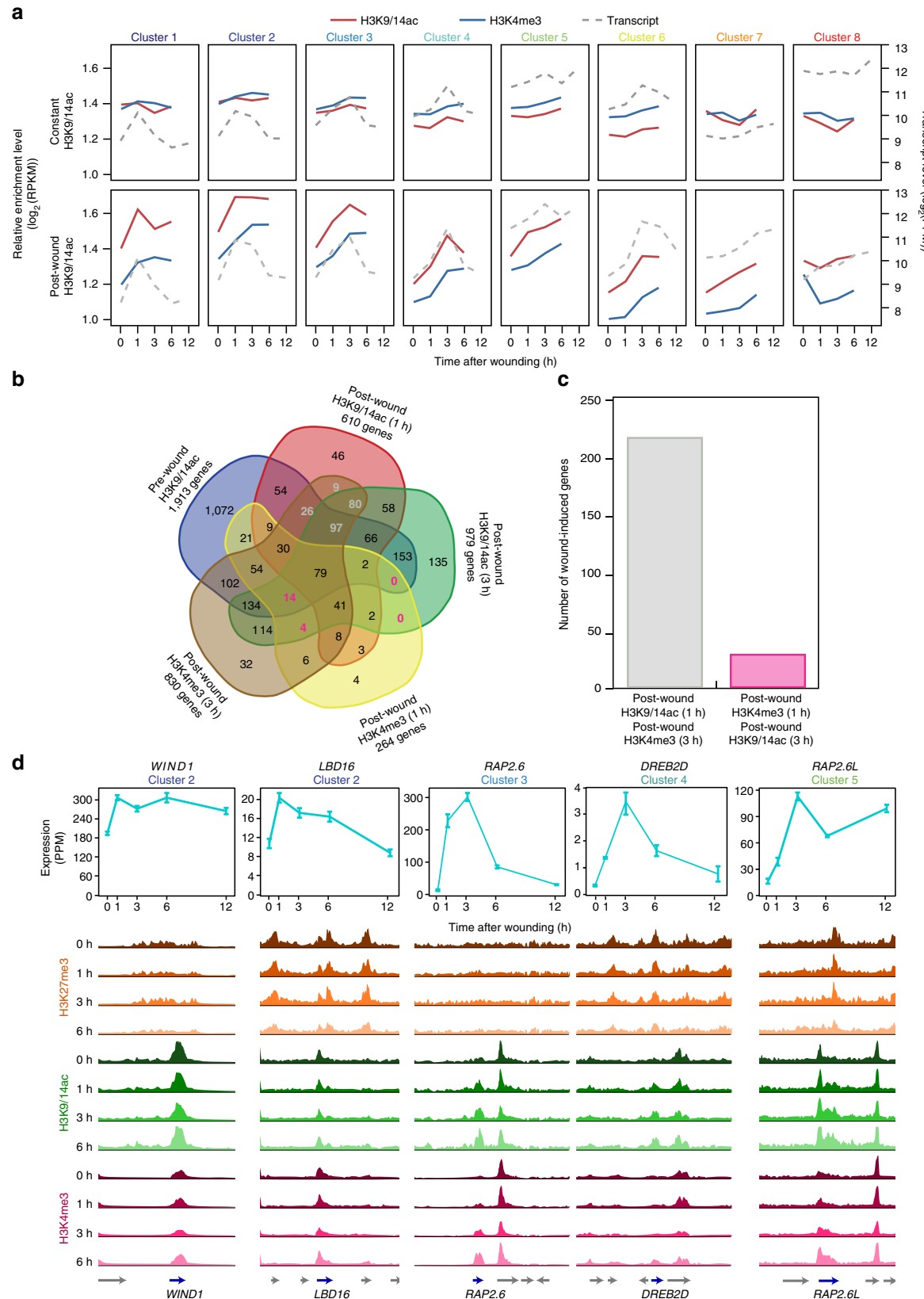

with H3K27me3 and still display similar wound-induced transcriptional response as *WIND1* and *RAP2.6*, on which H3K27me3 accumulation is negligible (Fig. 5d). We should also note that the transient character of early wound-induced transcription is not tightly coupled with relatively stable changes in histone modification states, as both H3K9/14ac and H3K4me3 levels remain high after transcript levels drop back to pre-wound levels (Fig. 5a, d). Similarly, we found that many wound-induced genes involved in hormone synthesis or signaling are marked with H3K9/14ac and H3K4me3 before wounding or gain these

**Fig. 5** Increase of H3K9/14ac marking precedes or coincides with H3K4me3 deposition and transcriptional activation after wounding. **a** Temporal dynamics of H3K9/14ac and H3K4me3 enrichment levels for wound-induced genes in the transcriptionally induced clusters 1–8. Average levels of H3K9/14ac and H3K4me3 are shown for the 1149 genes with constant H3K9/14ac (top panel) and the 1319 genes gaining post-wound H3K9/14ac (bottom panel). Transcriptional changes of the corresponding genes are plotted for comparison. **b** Venn diagram comparing the timing of pre- or post-wound deposit of H3K9/14ac and H3K4me3. Colored numbers indicate genes that gain H3K9/14ac before H3K4me3 (gray), and that gain H3K4me3 before H3K9/14ac (pink). **c** Total numbers of genes that gain H3K9/14ac before H3K4me3 (gray) and that gain H3K4me3 before H3K9/14ac (pink). **d** Integrative Genomics Viewer traces of H3K27me3, H3K9/14ac, and H3K4me3 enrichment at 0, 1, 3, and 6 h after wounding for *WIND1*, *LBD16*, *RAP2.6*, *DREB2D*, and *RAP2.6 L* genes. Blue arrows indicate the directionality of transcription. Line graphs display the transcriptional changes after wounding and error bars indicate the standard deviation (*n* = 3 independent experiments)

marks after wounding (Supplementary Fig. 10). Some of these genes, such as *LIPOXYNEASE 3* (*LOX3*) and *NAC DOMAIN CONTAINING PROTEIN 3* (*NAC3*), are concomitantly marked with H3K27me3, which again does not seem to impede wound-induced transcriptional activation.

**GNAT-MYST histone acetyltransferases regulate wound response**. Given the strong correlation between elevated H3 acetylation levels and wound-induced transcriptional activation, we tested whether the inhibition of histone acetylation interferes with wound-induced callogenesis. We applied two inhibitors of histone acetyltransferase activity, γ-butyrolactone (MB3) which affects members of the GNAT-MYST family[60] and C646 which affects the CBP family[61]. We examined the effect of these inhibitors on callus formation from wounded hypocotyls, as calli are more consistently produced from hypocotyls than roots. As shown in Fig. 6a and Supplementary Data 8, plants subjected to 25–100 μM MB3 treatment display significantly reduced capacity to develop callus upon wounding. Moreover, calli produced from hypocotyls exposed to MB3 are significantly smaller (Fig. 6b, c, Supplementary Data 8), indicating that GNAT-MYST-family histone acetyltransferase is critical for wound-induced callus formation. In contrast, application of 100 μM C646 does not prevent callus induction at wound sites (Supplementary Fig. 11a, Supplementary Data 8), suggesting that CBP-family histone acetyltransferases are not involved. Among the 5 members constituting the GNAT-MYST family histone acetyltransferases, we further found that *HAG1*/*GCN5* and *HAG3* are required for wound-induced reprogramming, as mutations in these genes strongly interfere with callus formation at wound sites (Fig. 6d–f, Supplementary Data 8). The impact of mutations in genes coding for other members of GNAT-MYST family histone acetyltransferases, i.e. *HAG2*, *HISTONE ACETYLTRANSFERASE OF THE MYST FAMILY 1* (*HAM1*) and *HAM3*, is milder, if not negligible (Supplementary Figs 11b-d, Supplementary Data 8).

To assess whether these phenotypic defects are associated with alteration of histone acetylation, we performed ChIP-seq on plants exposed to 100 μM MB3 from 24 h before wounding and harvested at 0, 3, and 6 h after wounding. Since overall levels of H3K9/14ac are not reduced to a level detectable by western blot (Supplementary Figs. 11e, 12), we normalized its enrichment to histone H3 levels. As expected, however, relative H3K9/14ac levels are reduced for a large proportion of genes in MB3-treated plants (Fig. 7a, Supplementary Data 6). For comparison, we also examined the enrichment of H3K4me3 in MB3-treated plants and found that their levels are also affected for 40–50% of genes presumably as a consequence of dampening of H3 acetylation (Fig. 7a, Supplementary Data 6). To test whether these changes in histone acetylation levels modifies the post-wound expression dynamics, we performed RNA-seq and found 4957 genes significantly down-regulated in MB3-treated plants compared to non-treated plants (Fig. 7b). Importantly, MB3 suppresses 1392 genes out of 3665 wound-induced genes and among them, 1024 genes show reduction of H3K9/14ac levels by MB3 (Fig. 7b,

Supplementary Data 6). These data thus demonstrate that GNAT-MYST-mediated histone acetylation is a central regulator of wound-induced transcriptional induction, and that it controls the transcription both directly and indirectly.

To explore how MB3 affects post-wound transcriptional dynamics over time, we plotted the average expression levels for wound-induced genes in the expression clusters 1–8. We found that MB3 treatment lowers the expression levels over time in each cluster (Fig. 7c). Consequently, the transcriptional maxima of early, transiently activated genes, i.e. those in clusters 1–4, are strongly reduced, and genes that are activated more gradually, i.e. those in clusters 5–8, are globally down-regulated over time. MB3 exposure strongly attenuates the expression of reprogramming or wound-response genes that are acetylated before and/or after wounding, causing overall down-regulation for some, such as *WIND1*, *LBD16*, *RAP2.6*, and *DREB2D*, and delayed transcriptional activation for others, such as *RAP2.6L* (Fig. 7d). For four of these genes (*WIND1*, *RAP2.6*, *DREB2D*, *RAP2.6L*), alteration of their expression in MB3-treated plants correlates with reduced H3K9/14ac levels (Fig. 7d). We should note, however, that these reductions are relatively mild after our MB3 treatment, which is consistent with the western blot data that did not detect visible reduction in H3K9/14ac levels. For *LBD16*, we did not observe clear reduction of histone acetylation in MB3-treated plants (Fig. 7d), implying that its wound-induced transcriptional activation is independent of GNAT-MYST-mediated histone acetylation. Nevertheless, these results altogether unveil that GNAT-MYST-mediated histone acetylation is crucial for defining both the level and timing of expression for wound-induced genes.

**Discussion**
Our assessment of histone modification states sheds new light on epigenetic regulation associated with early transcriptional induction during wound-induced cellular reprogramming. Particularly, our analysis highlights a prominent role for histone acetylation in facilitating transcriptional response upon wounding. The inhibition of GNAT-MYST-family histone acetyltransferases blocks wound-induced callogenesis, and compromised induction of several reprogramming genes, such as *WIND1*, *LBD16*, and *RAP2.6L*, accompanies this defect (Figs. 6, 7). Our data further showed that HAG1/GCN5 and HAG3 are the histone acetyltransferases involved in wound-induced callus formation (Fig. 6). A recent study, showed that HAG1/GCN5-dependent histone acetylation is necessary for in vitro shoot regeneration[50], by acetylating histones of several key root meristem genes, thereby activating their expression in hormone-induced callus. Our study thus together demonstrates a role for GNAT-MYST-mediated histone acetylation in both wound- and hormone-induced regeneration. Interestingly, histone acetylation promotes wound-induced callus formation (Fig. 6) but inhibits hormone-induced callus formation under in vitro conditions[50]. Thus, it appears that histone acetylation is not generally associated with onset of cell proliferation and callus formation, and that it is instead required for specific upstream processes during wound-induced callus formation. Previous studies have shown that histone acetylation is more

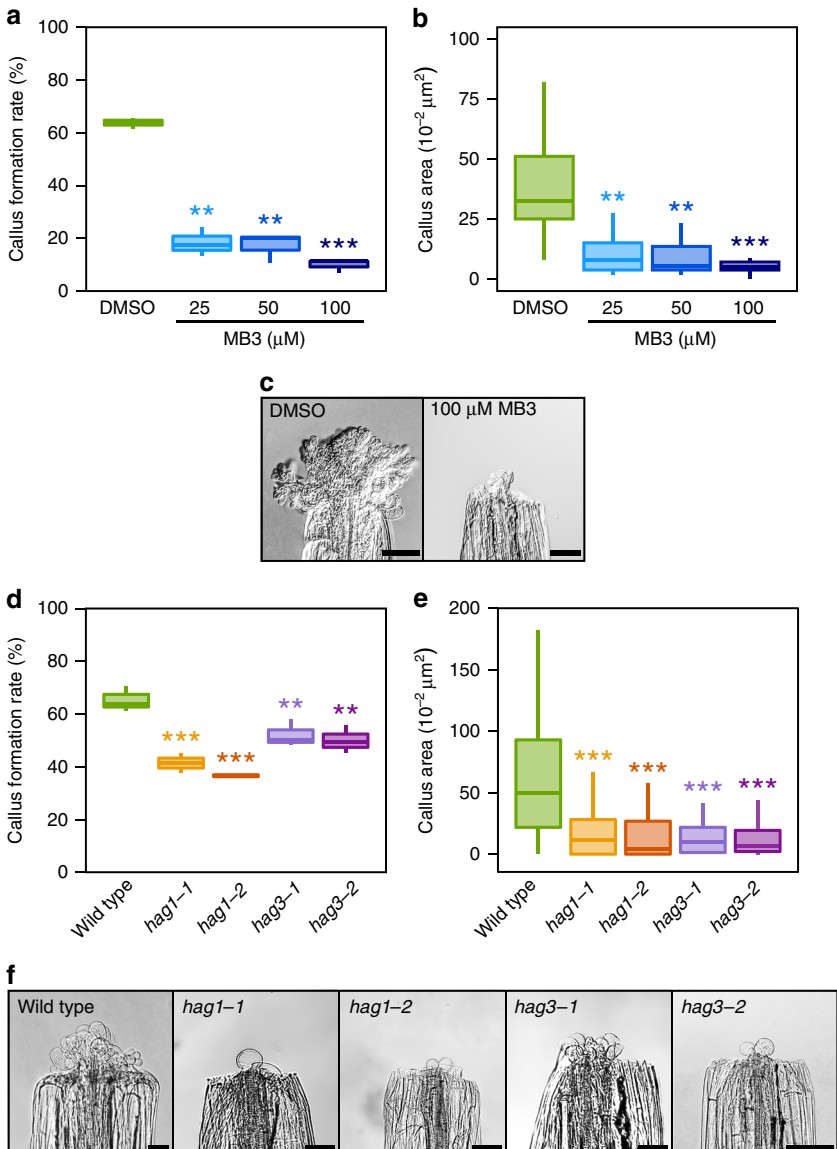

**Fig. 6** Inhibition of GNAT-MYST-mediated histone acetyltransferases impedes wound-induced callus formation. **a** Percentage of Arabidopsis hypocotyls that form callus at wound sites. Plants were treated with DMSO or 25–100 μM MB3 from 24 h before wounding. More than 40 plants were evaluated per experiment and error bars show the standard deviation ($n = 4$ independent experiments). **$p < 0.01$, ***$p < 0.001$ (Student's $t$ test). **b** Quantitative analysis of callus area in Arabidopsis hypocotyls treated with DMSO or 25 to 100 μM MB3. Error bars show the standard deviation ($n = 24$ biologically independent samples for DMSO-treated plants, $n = 17$ for MB3-treated plants). **$p < 0.01$, ***$p < 0.001$(Student's $t$ test). **c** Wound-induced callus in Arabidopsis hypocotyls treated with DMSO (left) or 100 μM MB3 (right). Scale bars = 100 μm. **d** Percentage of hypocotyls that form callus at wound sites in wild type, hag1-1, hag1-2, hag3-1, and hag3-2. More than 40 plants were evaluated per experiment and error bars show the standard deviation ($n = 4$ independent experiments). **$p < 0.01$, ***$p < 0.001$ (Student's $t$ test). **e** Quantitative analysis of callus area in wild type, hag1-1, hag1-2, hag3-1 and hag3-2 hypocotyls. Error bars show the standard deviation ($n > 100$ biologically independent samples). ***$p < 0.001$, (Student's $t$ test). **f** Wound-induced callus in wild type, hag1-1, hag1-2, hag3-1, and hag3-2 hypocotyls. Scale bars = 100 μm

dynamic than methylation[62] and it is involved in various stress and hormonal responses[63–67]. Our data therefore support the view that acetylation-mediated transcriptional modification is a general mechanism to swiftly respond to environmental or endogenous cues.

We should note that a substantial number of rapidly wound-induced genes are already marked with H3K9/14ac and H3K27ac before wounding, and that their acetylation levels are maintained after wounding (Fig. 5a). Therefore, high acetylation levels at these loci alone are not sufficient for the rapid transcriptional activation after wounding, and it is likely they act as a facilitator. This hypothesis is in agreement with the previously proposed model that

histone acetylation provides a more open, permissive chromatin state, but that the induction of gene expression per se requires the recruitment of transcriptional regulators to target loci[68]. Real-time live imaging in eukaryotic cells indeed showed that artificially induced H3K27ac accumulation enhances the binding of transcription factors to target promoters and thereby increases RNA polymerase II-mediated transcription[69]. It is thus likely that wound-induced transcriptional activation requires both histone acetylation and binding of wound-activated transcriptional regulators to target loci. Identifying these early acting transcriptional regulators is an important task in future studies to uncover the molecular mechanisms underlying wound-induced responses.

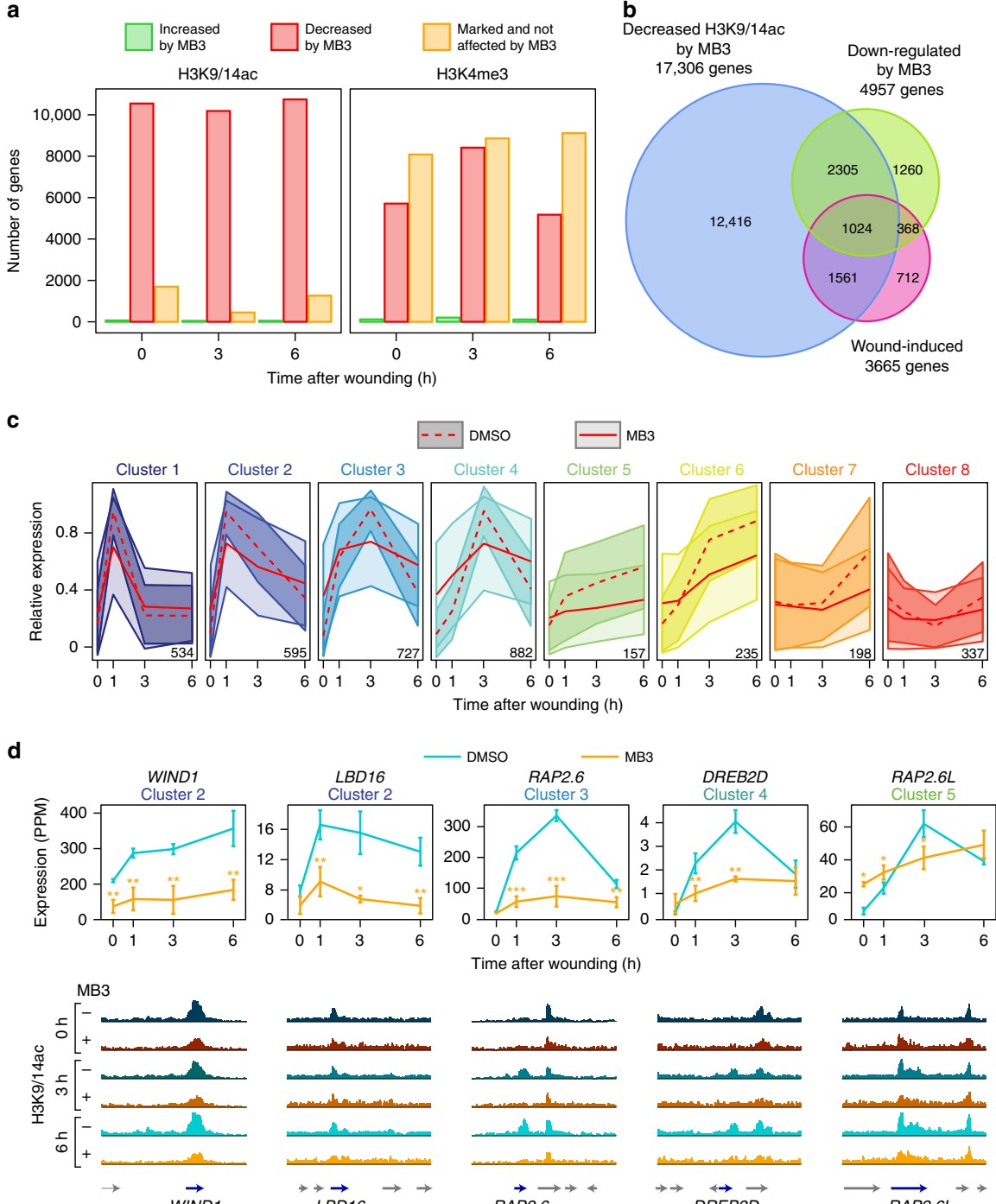

**Fig. 7** Inhibition of GNAT-MYST-mediated histone acetyltransferases interferes with wound-induced transcriptional activation. **a** Distribution of genes that gain or lose H3K9/14ac or H3K4me3 in plants treated with 100 μM MB3. Differentially marked genes include those that show marking changes with $\log_2(FC) > |0.15|$. **b** Venn diagram representation of the overlap between wound-induced genes, genes down-regulated after MB3 exposure and genes with decreased levels of H3K9/14ac after MB3 exposure. Genes down-regulated by MB3 correspond to those showing transcriptional down-regulation (FC < −1.5 and FDR (edgeR test) < 0.001) in plants treated with 100 μM MB3 compared to plants treated with DMSO, at one or multiple time points (0, 1, 3 and 6 h) upon wounding. Genes with decreased H3K9/14ac after MB3 exposure correspond to those that show changes in H3K9/14ac levels ($\log_2(FC) > |0.15|$) in plants treated with 100 μM MB3 compared to plants treated with DMSO, at one or multiple time points (0, 3, and 6 h) upon wounding. Intersection between wound-induced genes and genes down-regulated by MB3 (representation factor = 2.6 and p value < $3.3 \times 10^{-313}$). Intersection between the overlap described above and genes with decreased H3K9/14ac after MB3 exposure (representation factor = 1.5 and p value < $1.2 \times 10^{-71}$). **c** Ribbon plots displaying the wound-induced expression dynamics for genes in clusters 1–8 after exposure to DMSO or MB3. Red lines show the mean expression in plants treated with DMSO or 100 μM MB3, and the ribbons width represents the standard deviation (n = 3 independent experiments). **d** Integrative Genomics Viewer traces of H3K9/14ac enrichment at 0, 3, and 6 h after wounding in plants treated with DMSO (−) or 100 μM MB3 (+) for *WIND1, LBD16, RAP2.6, DREB2D,* and *RAP2.6 L* genes. Blue arrows indicate the directionality of transcription. Line graphs display the transcriptional changes after wounding in plants treated in the same conditions and error bars indicate the standard deviation (n = 3 independent experiments). *p < 0.05, **p < 0.01, ***p < 0.001 (Student's t test)

In contrast to histone acetylation, other histone modifications generally associated with active transcription, such as H3K4me3 and H3K36me3, do not appear to play major roles in post-wound transcriptional activation. Although we did observe an increase in H3K4me3 coupled with wound-induced transcription, its accumulation often peaks after transcriptional initiation and tends to follow H3 acetylation, particularly for rapidly induced genes (Fig. 5a). These observations suggest that H3K4me3 deposition is a downstream consequence of histone acetylation and resulting transcriptional activation. We observed few changes in H3K36me3 levels after wounding, and hardly any are associated with wound-induced transcription (Fig. 3). H3K36me3 is thought to participate in the regulation of transcriptional elongation rather than initiation[39], and consistently, the levels of H3K36me3 correlate with ongoing transcription in our dataset (Supplementary Fig. 2c). It is therefore likely that H3K36me3 is not involved in modifying the chromatin state to initiate gene expression upon wounding.

Interestingly, we also do not observe a strong correlation between gene activation and H3K27me3 dynamics. At least up to 6 h after wounding, H3K27me3 levels do not clearly drop at wound-induced loci (Fig. 3), suggesting a massive loss of H3K27me3 is not needed to induce PRC2 targets after wounding. Our data imply instead that genes co-marked with H3K27me3 and active marks are more likely wound-inducible. Whether these histone modifications actually co-occur within the same cell should be assessed by performing sequential ChIP in the future[70], as these marks might be present in different cells among the heterogenous population we sampled. It is, however, possible that the inducibility of gene expression is determined based on the balance between H3K27me3 and active marks, which may collectively influence chromatin accessibility. Interestingly, a recent study also reported a lack of correlation between changes in H3K27me3 levels and gene expression during the transition to flowering[52]. In line with this observation, Buzas et al.[71] showed that H3K27me3 is removed only after the transcriptional initiation of FLC. Complete loss of H3K27me3 may thus be unnecessary for the initial transcriptional activation of a PRC2 target gene, but it might instead play a role later in time, for instance, to maintain a permissive chromatin state. To explore this hypothesis, it would be interesting to investigate the transcriptional responses and epigenetic dynamics over a longer time after wounding. Another plausible hypothesis is that H3K27me3 acts as a buffer to rapidly dampen the expression of certain wound-induced genes after initial transcriptional bursts. Examining the expression of wound-induced PRC2 target genes in prc2 mutants should clarify whether PRC2 activity is required to reset their gene expression. Alternatively, it is possible that a limited population of cells within wounded explants lose H3K27me3, making these changes undetectable in the mixed population of cells in our experiment. It is known that some PRC2 target genes, such as ERF115, are expressed very locally at wound sites[16]. While we could detect relatively sharp increase in their transcript levels, detecting the loss of H3K27me3 in this limited cell population is more challenging. Combining ChIP-seq with other techniques such as INTACT, which permits isolation of specific cell populations[72], may be helpful in exploring the wound-induced chromatin dynamics at better cellular resolution.

## Methods

**Plant materials and growth conditions**. A rabidopsis thaliana (ecotype Col-0) plants were grown on Murashige-Skoog (MS) media containing 1% sucrose and 0.6% (w/v) gelzan. The hag1-1 (SALK_150784), hag1-2 (SALK_03913), hag2 (SALK_051832), hag3-1 (GABI 555H06), hag3-2 (SALKseq_027726), ham1 (SALK_027726), ham2 (SALK_106046) lines were all in the Col-0 background and obtained from the Arabidopsis Biological Resource Center. Plants were aligned on MS plates overlaid with a nylon mesh (Sefar) and grown for 7 days under continuous light conditions at 22 °C. To apply wound stress, roots were cut into 3 mm explants and harvested at 0, 1, 3, 6, and/or 12 h after wounding. To induce callus from hypocotyls, 7-day-old etiolated hypocotyls were cut at 7 mm from the hypocotyl-root junction

and incubated at 22 °C in the dark. Callus formation was quantified after 4 days as the percentage of explants with more than one callus cells developing from the apical end of hypocotyls. The projected callus area was quantified at 7 days after wounding using ImageJ. γ-butyrolactone (MB3, CAS No. 778649-18-6, ab141255, Abcam) and C646 (CAS No. 328968-36-1, Sigma) were applied 24 h before cutting by transferring the nylon mesh to MB3- or C646-containing MS plates.

**RNA sequencing and data analysis**. Total RNA was isolated in biological triplicates using the RNeasy plant mini kit (Qiagen). Isolated RNA was subjected to library preparation with the Kapa stranded mRNA sequencing kit (KK8420, Kapa Biosystems) and Illumina-compatible FastGene adapters (NGSAD24, Nippon Genetics). Single-end sequencing was performed on an Illumina NextSeq500 platform, and mapping was carried out using Bowtie[73]. For each RNA-seq experiment, over 80% of the reads were uniquely mapped to the TAIR10 Arabidopsis genome, resulting in 8–35 million mapped reads per sample. Differentially expressed genes were identified using the edgeR package on R/Bioconductor[74]. To retrieve wound-induced genes, transcript levels at each time point after wounding were compared to those prior to wounding. To retrieve genes differentially expressed after exposure to MB3, transcript levels between control and MB3-treated plants were compared at each time point after wounding. Differentially expressed genes were defined as those that show more than 1.5-fold changes in transcript levels at any time point with a false discovery rate (FDR) lower than 0.001. Gene clustering analysis was performed using K-Means from the MEV software after normalizing and mean centering the data. Graphical representation of the data was performed using the following R-packages: ggplot2, upsetr, eulerr, ggpubr, VennDiagram, and cowplot.

**ChIP sequencing and data analysis**. ChIP was carried out in duplicates as previously reported[75] with minor modifications. Around 1 g of 3-mm root explants was harvested and frozen using liquid nitrogen. Samples were ground to a fine powder using a multi-beads shocker (MB1200, Yasui Kikai) and the nuclear fraction was isolated after cross-linking for 10 min using 1% formaldehyde (Sigma) under vacuum. Chromatin was sheared at 4–6 °C for 15 min with a focused ultrasonicator (Covaris) with the following settings: duty cycle 5%, intensity 4, and cycles per burst 200. Sheared chromatin was immunoprecipitated using antibodies against histone H3 (ab1791; Abcam), H3K27me3 (07-449; Millipore), H3K4me3 (ab8580; Abcam), H3K36me3 (ab9050; Abcam), H3K9/14ac (06–599, Millipore), and H3K27ac (ab4729, Abcam). The isolated DNA was quantified with the Qubit dsDNA High Sensitivity Assay kit (Thermo Fisher Scientific), and 1–5 ng of DNA was used to make each ChIP-seq library. Libraries were prepared using the KAPA Hyper Prep Kit for Illumina (KK8502, KAPA Biosystems) and Illumina compatible adaptors (E7335, E7500, E7710, E7730, NEB). The 300–500 bp DNA fragments were enriched using Agencourt AMPure XP (Beckman Coulter). Libraries were pooled and 50-bp, single-read sequences were obtained with Illumina NextSeq500 sequencer at a depth of at least 10 million mapped reads.

Reads were mapped to the TAIR10 Arabidopsis genome using Bowtie[73], keeping only the reads mapped to a single unique position. To detect peaks, the default MACS2[76] settings were used for H3K4me3, H3K9/14ac, and H3K27ac, and the "−broad" option was used for H3K36me3 and H3K27me3. Histone H3 peaks were used as control for the baseline, and only peaks with $Q < 0.001$ were considered. Significant peaks were mapped to genes with BEDTools[77] using the "closest" function. Only the genes associated with a peak in both replicates were considered for further analysis. Statistical over- and under-representation of genes with different combinations of histone marks among the wound-induced genes was calculated based on hypergeometric statistical tests. The number of reads for each modification as well as histone H3 was averaged for each gene on an interval covering the 1 kb promoter region and the gene body, using the "coverage" function from BEDTools. To obtain the level of modification relative to histone H3 levels, the reads for each modification were normalized for histone H3 coverage within the same region using R. The distribution of histone modification levels was compared between wound-induced and non-induced genes, using density plots and Wilcoxon statistical tests. The level of modification at different points after wounding was compared by applying the MAnorm normalization[78] in R and differentially marked genes were selected based on a M-value < 0.15 for H3K27me3 and > 0.15 for other marks. Statistical analysis and graphical representation of the data were performed using the factoMiner, faxctoextra, ggsignif, ggplot2, superheat, and ggpubr R-packages. For principal component analysis of the marking level before wounding, the "pca" and "fviz_pca_biplot" function of the R packages "factoextra" and "FactoMineR" were used. The following variables, ranking of histone modification level for each mark, presence in cluster 1–8 as depicted in Fig. 2, and ranking of expression level before wounding, were considered for each gene. To visualize peaks, bigwig files that were averaged over the two replicates and normalized to 10 million reads were created using the deeptools package[79] and displayed using Integrative Genomics Viewer (version 2.3.80, Broad institute).

**Western blot analysis**. Western blot analysis was performed in duplicate. Chromatin was extracted from around 50 mg of 3 mm root explants using a salt extraction method[80] with some modifications. Samples were ground using a multi-beads shocker (MB1299, Yasui Kikai) and homogenized in extraction buffer (10 mM HEPES pH 7.9, 10 mM KCl, 1.5 mM $MgCl_2$, 0.34 M sucrose, 10% glycerol,

0.2% NP-40, 1 mM PMSF and 10 mM sodium butylate). After incubation on ice for 10 min with gentle mixing, samples were centrifuged at 6500$g$ for 5 min at 4 ℃. The pellet was resuspended into extraction buffer without NP-40, incubated for 1 min on ice and centrifuged. The pelleted nuclei were lysed by vortexing in a no-salt buffer (3 mM EDTA, 0.2 mM EGTA, 1 mM PMSF, 10 mM sodium butylate) and incubated for 30 min on ice with gentle mixing every 5 min. The samples were then centrifuged at 16,000$g$ for 10 min at 4 °C, and the pelleted chromatin was dissolved into a nuclei storage buffer (20 mM HEPES pH 7.9, 5 mM MgCl$_2$, 20% glycerol and 1 mM DTT)[81]. The total amount of extracted proteins was quantified with the Qubit dsDNA High Sensitivity Assay Kit, and 1 µg of protein was used for SDS-PAGE. After transfer to a PVDF membrane, histone H3 and its modifications were detected with the same set of antibodies as those used for ChIP seq.

**GO enrichment analysis**. The agriGO tool[82] was used to retrieve the GO terms significantly enriched ($p < 0.01$) among the clusters of genes showing an altered timing of response in the RNA-seq datasets. The top 10 most significant GO terms for each cluster was then compared for common terms. In order to evaluate GO term enrichment among loci associated with different chromatin states before wounding, we compared the ratio of marked genes associated with selected GO terms to the prevalence of these GO terms in the whole genome via a hypergeometric test. Gene lists associated with each selected GO terms were retrieved from TAIR10.

**Statistics and reproducibility**. For RNA seq, ChIP seq and western blot analysis, pools of roots were used as biological replicates. Pools were obtained from >300 plants grown on different petridishes that were randomly placed in a growth chamber to avoid petridish effects.

RNA seq was performed in biological triplicate. EdgeR algorithm was used to define the significant transcripts, for each time point after wounding a contrast with the pre-wounding data point was considered. For MB3 experiments the contrasts were between with or without treatment.

ChIP seq for histone marks after wounding was performed in biological duplicate. Only peaks that were discovered above the background in both replicates were considered for further analysis. ChIP seq for MB3 marks was performed as a single replicate at three time points that showed comparable results.

Western blot analysis was performed in duplicate. In both replicate we found no obvious changes in intensity. Therefore, only one of the replicates is shown in the figures.

For phenotypic analysis, we counted callus formation ratio and size in four independent experiments each time using >40 individual plants.

**Reporting summary**. Further information on research design is available in the Nature Research Reporting Summary linked to this article.

## Data availability

RNA-seq and ChIP-seq data generated in this study have been deposited to ArrayExpress with accession numbers E-MTAB-7609, E-MTAB-7611, E-MTAB-7612 and E-MTAB-8265.

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

## Acknowledgements

The authors are grateful to the members of Sugimoto's lab for discussion and Taiko To for her advice on western blot analysis. This work was supported by a grant from RIKEN to B.R. (201701100428), a grant from MEXT to K.S. (15H05961), a grant from JSPS to K.S. (17H03704), a PHC SAKURA grant to K.S. and F.R. (No 36164XF) and a PICS grant to F.R. (CNRS-INSB No 7625). A.L. is supported by a fellowship from the University of Tokyo and M.I. is supported by a fellowship from JSPS.

## Author contributions

B.R. and K. Sugimoto designed research with inputs from A.I., M.I., A.L., and F.R. B.R. and A.K. performed the RNA-seq and ChIP-seq experiments and B.R. and A.L. analyzed the data with help from S.I. and T.K. T.S. sequenced RNA-seq and ChIP-seq libraries. A.T., A.I. D.F., and Y.S. performed western blot experiments. K. Sako and M.S. provided Arabidopsis mutants. B.R., A.L., and K. Sugimoto wrote the manuscript with inputs from all the authors.

## Competing interests

The authors declare no competing interests.
