## [Peer Review File · Communications Biology]

Reviewers' comments:

Reviewer #1 (Remarks to the Author):

In this study the Authors survey changes in transcription at different times after wounding. This extends previous findings. Then they compare the set of genes affected by wounding with marks in WT. They observe that marks are correlated with their associations revealed in previous reports. As expected, genes transcriptional induced by wounding tend to associate with K27me3 before wounding and also with active marks associated with potential expression but not K36me3 associated with elongation.

Then levels of the same marks are profiles after wounding at different time points. They report a stronger correlation of transcriptional activation with accumulation of K9/14 Ac and K4me3 and that deposition of these two marks follows a distinct temporal pattern with acetylation often preceding methylation

The authors then apply inhibitors of acetyltransferases and observe less callus induction after wound. They correlate the impact of inhibitor treatment with transcription profiles.

The study confirms to a large extent our current knowledge of the link between histone modifications studied and transcription in animal cells and the study suffers from several major problems in its experimental design as detailed below.

1. Compared with similar studies performed with cell lines, a major problem with the study is the heterogeneity of the tissue. It is unclear whether all cells express wound induced genes and from which cells the chromatin analyzed comes from. But this is currently difficult to solve. Yet this prevents strong correlative conclusion since the various changes observed could occur in different cells.
2. What is more problematic is the lack of spike-in that is required to reach quantitative measurements. Without spike-in it is not possible to compare absolute levels of enrichment between samples. The data presented here thus provides trends and this prevents many of the strong conclusions presented in the discussion and summary.
3. The major novel conclusion of the study "These data thus demonstrate that GNAT MYST-mediated histone acetylation is a central regulator of wound-induced transcriptional induction and that it affects the transcriptional response both directly and indirectly." is not supported. The authors do not monitor the effect of inhibitors on levels of acetylation marks. They only monitor transcription and make association with the presence of the marks of interest in absence of treatment. It is important that the authors profile acetylation marks and at least one or two methylation marks as controls from tissue of treated plants before and after wounding.

Reviewer #2 (Remarks to the Author):

Review comm. Biol.

In their work, Ryman et al., provide detailed view of the genome-wide dynamics of selected histone modifications after wounding stress in Arabidopsis, and correlate these chromatin changes with

transcriptional response. They show that the H3 K9/14 and H3K27 acetylation are common marks of wound-induced, and for most genes are deposited before stress and then their levels increase after wounding. Crucial role of H3 acetylation is further supported by showing that inhibition of the GNAT-MYST histone acetyltransferases (MB3 inhibitor) affects wound-induced callus formation and transcriptional activation of wound-induced genes. In general, the manuscript brings new and relevant results to the field and shows that chromatin modifications are important components of plant responses to stress. On the other hand, major weakness of the manuscript is correlative nature of the analyses which are not very well supported by genetic and physiological analyses. For example, functional analysis of GNAT-MYST acetyltransferases would strengthen the conclusions drawn from genomic analyses and improve the manuscript. In addition, some parts of the manuscript need better explanation. Here are my specific comments:

1) This work relates to the previous one by the same group (Ikeuchi et al., *Plant Phys.*, 2017) and has similar design. The largest difference is that different tissues were used (hypocotyls in the previous work, roots in the current work). It would be more convenient if the same plant tissue was used, and it is not clear why the authors chose to work on different one. Even if there were only technical issues like the amount of material needed, it should be specified in the manuscript.

2) Only wound-induced genes are studied. I suppose authors wanted to focus on these genes to correlate transcriptional changes with histone acetylation. However, in my opinion the information about wound – repressed genes is also relevant for this work. Do these genes fall into ‘non-induced’ category on fig. S3b? How many repressed genes were detected? Do they gain H3K27me or loose acetylation marks ?

3) The authors point to underrepresentation of H3K36me3 among wound induced-genes (p.4). This result has been left without the interpretation

4) The Authors noted that the genes co-marked with H3K27me3 and active marks are overrepresented among wound-induced genes (p.4, p.7). However, as experiments were performed on mixture of cells from different tissues from different plants, the observed effect can be attributed to the different marks being deposited in those different cells. This should be at least noted in the text, or the Authors can try to address this experimentally.

5) In their previous work much attention was paid to the hormonal pathways that are important for wound response in hypocotyls. It is a little bit weird that they do not show any hormone-related genes in the current work but only mention them in Discussion.

6) Venn diagrams on the figures are very hard to interpret. I would suggest reduction of the data (for example H3K36me that was found to be not correlated with transcriptional response) and moving some part to supplementary materials.

7) Information about clusters to which the genes shown on fig. 5d and 7c belong to should be provided.

8) The experiment with HATs inhibitors is very important for the paper as it shows relevance of histone acetylation in response to wounding. These results could be strengthened by analysis of wound response in GNAT-MYST acetyltransferase mutants (e.g. *gcn5/hag1*).

9) Fig. 7c shows expression changes of selected genes after wounding and MB3 treatment, but the levels of H3 acetylation after the inhibition should be also shown.

10) Some figures are not enough described in the text and/or figure legends and therefore difficult to interpret and follow the authors’ conclusions: Fig.1c , how to interpret percent numbers and representation factor? , Fig. 2b , description of how the PCA was performed is not clear, and to me the biggest difference is between H3K27me3 and H3K36me3

Minor points:

- the kinetics of expression changes of some genes shown in fig. S1c is different in roots and hypocotyls and it should be mentioned in the text.

- P.5, line 168 : the Authors comment on early induced genes (3h) but what about earlier time points ?

- P. 7, please specify cluster numbers when Fig. 5Aa is mentioned
- description of GO analysis on histone modifications (methods) is not clear

Reviewer #3 (Remarks to the Author):

In this study, Rymen and colleagues examine the changes in gene expression and chromatin state that occur after wounding in *A. thaliana* roots. Previous work had suggested that changes in chromatin state were important for regulating genes involved in the regenerative process, but it was unclear which marks played a role in this reprogramming or which factors were involved. The authors performed RNA-seq and ChIP-seq before wounding and at several timepoints post-wounding. They looked at several histone modifications that might play a role in regulation - H3K27me3, a repressive mark, and the activating marks H3K36me3, H3K4me3, H3K9/14ac and H3K27ac. The authors found that all of the acetylation marks tested, but none of the methylation marks, showed strong changes after wounding that correlated with gene expression changes. They also showed that inhibiting histone acetyltransferase activity with MB3 prevents callus formation and attenuates gene expression changes after wounding, consistent with histone acetylation playing an important role in regeneration after wounding. Surprisingly, H3K27me3 and H3K36me3 levels remained nearly constant at genes induced after wounding, suggesting these are not involved in wound response, while H3K4me3 patterns suggested that this mark may follow but not trigger the gene expression program associated with wounding.

Overall I felt that this was an interesting and well-presented paper with novel findings consistent with predictions drawn from previous work in epigenetic reprogramming and stress/hormone response in plants. The paper was well organized, easy to read, and generally very thorough and careful in its interpretation of the data. The results highlight several interesting new questions, and provide a starting point for future work to examine which histone acetyltransferases are involved in facilitating the wounding response, and how they recognize their target genes. I did have a few minor comments/suggestions, listed below.

1. I found Fig. 2B hard to interpret overall. I would suggest dropping 2B and expanding 2C to show the same panel three times: as-is, sorted by H3K27me3, and sorted by H3K36me3. This will show that it's only the levels of pre-wounding acetyl marks that correlate well with the timing of induction. Expression at 0h could also be added as a column in those panels.
2. Fig. 4 shows only data for the 3665 wound-induced genes, which makes it hard to tell how unusual the distribution of the 8 categories shown is. It would be useful to add the same pie-chart for all (expressed) genes next to the one for the 3665 wound-induced genes in panel A. p-values could also be estimated, e.g. using a bootstrapping approach, for the number of genes in each of the 8 categories in panel A vs. what would be expected from a set of 3665 genes drawn at random. Those p-values could also be added to the text (line 226, etc.).
3. Fig. 1D - The heatmap is currently colored by p-value, but the magnitude of the p-value is not a good measure of effect size. Instead, the heatmap could be colored based on representation factor, with a scale indicating what a value > 1 indicates vs. a value < 1 (I had to google the meaning of 'representation factor' in the context of hypergeometric tests, despite having used those tests often in the past, so others may not be familiar with the term). This will also help differentiate pairs of marks that co-occur from pairs that are anticorrelated. Significant vs. nonsignificant cells could be indicated using significance stars, for example.

4. In Fig. 3D, y-axis refers to fold-change (FC). I assume it's indicating FC of induction after wounding relative to pre-wounding mark levels, but please indicate explicitly.

5. Another hypothesis for why H3K27me3 levels were unaffected at induced genes could be that marking by H3K27me3 helps rapidly turn some wound-induced genes 'off' after the initial burst, preventing erroneous transcription. An interesting future direction might be to look at induction of these genes in a PRC2 mutant, to see if wound-induced genes fail to get turned off in this background. Alternatively, contrasting the pattern of gene activation over the 6h time series for genes that gain acetylation and are marked by H3K27me3, vs those that gain acetylation but have no H3K27me3, might be interesting.

6. It is interesting that loss of H3K27me3 was not required for upregulation of marked genes or gain of acetylation. H3K27me3 and H3K27ac are mutually exclusive, since (as far as I know) the lysine can't be simultaneously methylated and acetylated - this would suggest that genes that gain H3K27ac should lose H3K27me3, but this appears to not be the case in these data. What specifically was the relationship between these two marks in the data? Was H3K27ac only gained over genes with no H3K27me3? It might be useful to add the H3K27ac tracks to Fig. 5D.

7. My impression is that histone acetylation is generally less "stable" than methylation - turnover rates for methyl marks are much slower than for acetyl marks (for example, see Mews et al. 2014 Mol. Cell Biol. paper on histone methylation/acetylation on emerging from quiescence). So, methylation is often a more long-term, stable mark, whereas acetylation can be added and then removed quickly. Then, it is perhaps not surprising that all of the responsive marks detected in this study, which used a relatively short time series (6h), were acetyl marks. Perhaps in general, acetylation might be more likely to be involved in environmental responses that require sudden, rapid activation of a gene expression program. Are there any other studies that have shown a role for acetylation in mediating stress or hormone responses in plants in contexts other than regeneration?

We are grateful to the editors and referees for their thoughtful comments on our manuscript. We provide a point-by-point response explaining how we have addressed each of the reviewers' concerns and suggestions below.

Reviewer #1 (Remarks to the Author):

In this study the Authors survey changes in transcription at different times after wounding. This extends previous findings. Then they compare the set of genes affected by wounding with marks in WT. They observe that marks are correlated with their associations revealed in previous reports. As expected, genes transcriptional induced by wounding tend to associate with K27me3 before wounding and also with active marks associated with potential expression but not K36me3 associated with elongation.

Then levels of the same marks are profiles after wounding at different time points. They report a stronger correlation of transcriptional activation with accumulation of K9/14 Ac and K4me3 and that deposition of these two marks follows a distinct temporal pattern with acetylation often preceding methylation

The authors then apply inhibitors of acetyltransferases and observe less callus induction after wound. They correlate the impact of inhibitor treatment with transcription profiles.

The study confirms to a large extent our current knowledge of the link between histone modifications studied and transcription in animal cells and the study suffers from several major problems in its experimental design as detailed below.

We agree that some of the observations on the link between histone modification and transcription correlate with previous findings in animal cells. We need to emphasize, however, that this link has never been studied in the context of wound response or regeneration in both animals and plants, and thus we do provide novel mechanistic insights into how wound stress modulates the epigenetic landscape in order to promote cellular reprogramming. As discussed below, we have addressed the issues raised by the reviewer on our experimental designs.

1. Compared with similar studies performed with cell lines, a major problem with the study is the heterogeneity of the tissue. It is unclear whether all cells express wound induced genes and from which cells the chromatin analyzed comes from. But this is currently difficult to solve. Yet this prevents strong correlative conclusion since the various changes observed could occur in different cells.

We are aware that cellular heterogeneity is a potential issue. However, we aimed in this study to uncover the global wound-induced epigenetic changes that can still be observed within heterogeneous cell populations. The main correlative conclusion made in this manuscript is that histone acetylation is required for wound-induced cellular reprogramming, and we now provide data generated using a chemical inhibitor as well as mutants to support this claim further. As pointed out by reviewers 2 and 3, apparent co-occurrence of some histone marks might be attributable to the fact that we sample from heterogeneous cell populations. We thus added the following sentence to discuss this point in the "Discussion" section:

"Our data imply instead that genes co-marked with H3K27me3 and active marks are more likely to be wound-inducible. Whether these histone modifications actually co-occur within the same cell should be assessed by performing sequential ChIP analysis in the future⁷⁰, as these marks might be present in different cells among the heterogenous population we sampled."

2. What is more problematic is the lack of spike-in that is required to reach quantitative measurements. Without spike-in it is not possible to compare absolute levels of enrichment between

samples. The data presented here thus provides trends and this prevents many of the strong conclusions presented in the discussion and summary.

We agree with the reviewer that spike-in is essential when absolute levels of histone modification may vary to a large extent between samples, for instance when comparing between wild-type and mutants (Jiang and Berger, 2017; Nassrallah *et al.*, 2018). This is not the case, however, when absolute levels remain comparable between samples, and it is a standard practice, as found in many published epigenomic papers, to examine H3 modification enrichment relative to histone H3 levels (Liu *et al.*, 2016; Moreno-Romero *et al.*, 2019; See *et al.*, 2019; Xiao *et al.*, 2017; Yang *et al.*, 2018). Our new western blot data revealed that overall levels of histone marks remain relatively constant over time after wounding (Figure S6) and upon short exposure to MB3 (Figure S11e). These data thus support that our ChIP-seq datasets, also normalized against histone H3, are valid to discuss relative enrichment of histone marks. We added the following sentences in the “Results” section to clarify this point:

“Several recent studies used exogenous epigenome, *i.e.* chromatin from drosophila or human, as a reference to compare the level of histone modification between wild-type and mutants, as histone modification levels between these samples may differ to a large extent^{53,54}. When the total modification levels remain comparable between samples, however, many studies have examined H3 modification enrichment relative to histone H3 levels and successfully detected locus-specific alterations⁵⁵⁻⁵⁹. Given that overall levels of histone marks do not change drastically within our time points based on a western blot analysis (Figure S6), we normalized our ChIP-seq data for these marks to histone H3, and evaluated whether their relative enrichment levels change at specific loci compared to unwounded plants.”

We also added the following sentence in the “Methods” section to explicitly state that our ChIP-seq datasets are normalized against histone H3:

“The number of reads for each modification as well as histone H3 was averaged for each gene on an interval covering the 1 kb promoter region and the gene body, using the “coverage” function from BEDTools. To obtain the level of modification relative to histone H3 levels, the reads for each modification were normalized for histone H3 coverage within the same region using R.”

3. The major novel conclusion of the study “These data thus demonstrate that GNAT MYST-mediated histone acetylation is a central regulator of wound-induced transcriptional induction and that it affects the transcriptional response both directly and indirectly.” is not supported. The authors do not monitor the effect of inhibitors on levels of acetylation marks. They only monitor transcription and make association with the presence of the marks of interest in absence of treatment. It is important that the authors profile acetylation marks and at least one or two methylation marks as controls from tissue of treated plants before and after wounding.

We agree that this is an important point, and thus included new ChIP-seq data showing that MB3 does indeed inhibit histone acetylation (Figures 7a, 7d). We described these data with the following sentences in the “Results” section:

“To assess whether these phenotypic defects are indeed associated with alteration of histone acetylation, we performed ChIP-seq analysis on plants exposed to 100 μ M MB3 from 24 h before wounding and harvested at 0, 3 and 6 h after wounding. Since overall levels of H3K9/14ac are not reduced to the level detectable by western blot after the short-term exposure to MB3 (Figure S11e), we normalized its enrichment to histone H3 levels. As expected, however, MB3 reduces the H3K9/14ac relative enrichment level for a large proportion of genes that possess this mark in control plants (Figure 7a and Table S6). For comparison, we also examined the enrichment of H3K4me3 in

MB3-treated plants and found that their levels are also affected for 40 to 50% of genes presumably as a consequence of dampening of H3 acetylation (Figure 7a and Table S6).”

“For four of these genes (*WIND1*, *RAP2.6*, *DREB2D*, *RAP2.6L*), alteration of their expression dynamics in MB3-treated plants correlates with a reduction of their H3K9/14ac levels (Figure 7d). We should note, however, that these reductions are relatively mild after the short-term MB3 treatment, which is consistent with our western blot data that did not detect a visible reduction in the overall H3K9/14ac levels. For *LBD16*, we did not observe a clear reduction of histone acetylation in MB3-treated plants (Figure 7d), implying that its wound-induced transcriptional activation is independent from GNAT-MYST-mediated histone acetylation.”

Reviewer #2 (Remarks to the Author):

In their work, Rymen et al., provide detailed view of the genome-wide dynamics of selected histone modifications after wounding stress in Arabidopsis, and correlate these chromatin changes with transcriptional response. They show that the H3 K9/14 and H3K27 acetylation are common marks of wound-induced, and for most genes are deposited before stress and then their levels increase after wounding. Crucial role of H3 acetylation is further supported by showing that inhibition of the GNAT-MYST histone acetyltransferases (MB3 inhibitor) affects wound-induced callus formation and transcriptional activation of wound-induced genes. In general, the manuscript brings new and relevant results to the field and shows that chromatin modifications are important components of plant responses to stress. On the other hand, major weakness of the manuscript is correlative nature of the analyses which are not very well supported by genetic and physiological analyses. For example, functional analysis of GNAT-MYST acetyltransferases would strengthen the conclusions drawn from genomic analyses and improve the manuscript.

Following the suggestion from the reviewer, we provide new genetic data to show that mutations in *HAG1/GCN5* and *HAG3*, but not in *HAG2*, *HAM1* or *HAM2*, cause strong defects in wound-induced callus formation (Figures 6d-f, S11a-c). We described these results in the “Results” section as follows:

“Among the 5 members constituting the GNAT-MYST family of histone acetyltransferases, we further found that those encoded by *HAG1/GCN5* and *HAG3* are required for wound-induced reprogramming, as mutations in these genes strongly interfere with callus formation at wound sites (Figures 6d-f). On the contrary, the impact of mutations in the genes coding for other members of the GNAT-MYST family histone acetyltransferases, *i.e.* *HAG2*, *HISTONE ACETYLTRANSFERASE OF THE MYST FAMILY 1 (HAM1)* and *HAM3*, is far milder, if not negligible (Figures S11b-d).”

And discuss them in the “Discussion” section as follows:

“Our genetic analysis further showed that *HAG1/GCN5* and *HAG3* are the primary histone acetyltransferases within the GNAT-MYST family that are involved in wound-induced callus formation (Figure 6).”

In addition, some parts of the manuscript need better explanation. Here are my specific comments:

1) This work relates to the previous one by the same group (Ikeuchi et al., Plant Phys., 2017) and has similar design. The largest difference is that different tissues were used (hypocotyls in the previous work, roots in the current work). It would be more convenient if the same plant tissue was used, and it is not clear why the authors chose to work on different one. Even if there were only technical issues like the amount of material needed, it should be specified in the manuscript.

We indeed had to switch our analysis from hypocotyls to roots so that we could quickly collect sufficient wounded materials to perform ChIP-seq and RNA-seq in parallel. To state this clearly, we added the following sentence in “Results”:

“The use of roots, instead of hypocotyls as in our previous study⁶, allowed us to upscale our sampling capacity to a level sufficient to perform RNA-seq and ChIP-seq analyses in parallel.”

2) Only wound-induced genes are studied. I suppose authors wanted to focus on these genes to correlate transcriptional changes with histone acetylation. However, in my opinion the information about wound – repressed genes is also relevant for this work. Do these genes fall into ‘non-induced’ category on fig. S3b? How many repressed genes were detected? Do they gain H3K27me or loose acetylation marks ?

As suggested by the reviewer, we added new data on wound-repressed genes (Figures 1b-d, 3c, S1a, S1b, S4, S5, S7). We detected 6,010 genes repressed within 12 h after wounding and found that the transcriptional repression of these genes largely correlates with a loss of histone H3 acetylation and not with an increase in H3K27me3. We described these findings in the “Results” section as follows:

“Wound-induced transcriptional repression is coupled with decrease in H3K9/14ac, H3K27ac and H3K4me3 marking

Using these ChIP-seq and RNA-seq time-course data, we investigated how pre-wound and post-wound histone marks correlate with transcriptional repression triggered by wounding. Among the 6,010 genes repressed by wounding, we found that 4,612 genes are marked before wounding with at least one of the histone modifications we analyzed (Figure 1b, Table S2). As might be expected, genes marked with pre-wound H3K27me3 are underrepresented (Figure 1c) likely because many of these genes are not expressed before wounding (Figure S2c). In contrast, genes with pre-wound permissive marks are overrepresented among the wound-repressed genes (Figure 1c). Similar to the wound-induced genes, we detected a significant enrichment of co-marked genes within the wound-repressed genes, among which those marked with both H3K27me3 and the permissive marks are highly represented (Figure 1d). To test whether the relative levels of histone modification marks before wounding correlate with the timing of repression, we grouped the 6,010 wound-repressed genes into 8 clusters defined by their repression dynamics (Figure S7a, Table S1). We did not observe any obvious correlation between the pre-wound levels of any of the marks analyzed and the expression cluster to which the marked genes belong (Figure S7b). Moreover, GO enrichment analysis performed on each of these clusters revealed that they represent very different sets of biological functions (Figure S8), suggesting that neither the timing of wound-induced repression nor the functional class of the wound-repressed genes correlate with the pre-wound relative enrichment levels of the chromatin modifications we tested.

Furthermore, we detected post-wound alterations of histone mark levels on only 20% of the 6,010 wound-repressed genes (Table S5). Similar to the wound-induced genes, H3K9/14ac, H3K27ac and H3K4me3 marks are most responsive to wounding for the repressed genes, as 1,106 of them show a decrease in at least one of the H3K9/14ac, H3K27ac or H3K4me3 marks (representation factor = 3.3 and p value ≈ 0 , Figure 3c). In contrast, only 143 wound-repressed genes lose H3K36me3 upon wounding (representation factor = 1.6 and p value $< 3.5 \times 10^{-9}$), and 37 gain H3K27me3 (representation factor = 0.1 and p value $< 2.7 \times 10^{-83}$) (Figure 3c), confirming that post-wound modification of these two latter marks poorly correlate with the transcriptional changes occurring within the first 12h after wounding.”

3) The authors point to underrepresentation of H3K36me3 among wound induced-genes (p.4). This result has been left without the interpretation

We added the following sentence in the “Results” section:

“In contrast, we noticed that genes associated with pre-wound H3K36me3 are significantly underrepresented among wound-induced genes, presumably because these genes are already actively transcribed before wounding (Figure 1c)”

We have also discussed our interpretation of these data in the “Discussion” section as follows:

“We observed very few changes in H3K36me3 levels after wounding, and hardly any are associated with wound-induced transcription (Figure 3). H3K36me3 is thought to participate in the regulation of transcriptional elongation rather than initiation³⁹, and consistently, the levels of H3K36me3 correlate with ongoing transcription in our dataset (Figure S2c). It is therefore likely that this histone mark is not involved in modifying the chromatin state to initiate new gene expression upon wounding.”

4) The Authors noted that the genes co-marked with H3K27me3 and active marks are overrepresented among wound-induced genes (p.4, p.7). However, as experiments were performed on mixture of cells from different tissues from different plants, the observed effect can be attributed to the different marks being deposited in those different cells. This should be at least noted in the text, or the Authors can try to address this experimentally.

We agree that this is an important point. We stated this possibility more clearly in the “Discussion” section as follows:

“Whether these histone modifications actually co-occur within the same cell should be assessed by performing sequential ChIP analysis in the future⁷⁰, as these marks might be present in different cells among the heterogenous population we sampled.”

5) In their previous work much attention was paid to the hormonal pathways that are important for wound response in hypocotyls. It is a little bit weird that they do not show any hormone-related genes in the current work but only mention them in Discussion.

Following the suggestion from the reviewer, we provided new data showing wound-induced changes of histone modification for hormone biosynthesis and signalling genes (Figure S10). Similar to reprogramming-related genes, hormone-related genes already possess or gain H3K9/14 and H3K4me3 after wounding. We described these findings in the “Results” section as follows:

“Similarly, we found that many wound-induced genes known to be involved in hormone synthesis or signaling are marked with H3K9/14ac and H3K4me3 before wounding or gain these marks after wounding although we did not find a particular correlation between specific hormonal pathways and the chromatin landscape (Figure S10). Moreover, some of these genes, such as *LIPOXYNEASE 3* (*LOX3*) and *NAC DOMAIN CONTAINING PROTEIN 3* (*NAC3*), are concomitantly marked with H3K27me3, which again does not seem to impede their wound-induced transcriptional activation.”

6) Venn diagrams on the figures are very hard to interpret. I would suggest reduction of the data (for example H3K36me that was found to be not correlated with transcriptional response) and moving some part to supplementary materials.

We simplified some of the Venn diagrams in the main figures (Figures 1a, 1b, 7b), and moved the others to supplemental data (Figure S3a). We believe that these changes ease their interpretation.

7) Information about clusters to which the genes shown on fig. 5d and 7c belong to should be provided.

We added this information in these figures.

8) The experiment with HATs inhibitors is very important for the paper as it shows relevance of histone acetylation in response to wounding. These results could be strengthened by analysis of wound response in GNAT-MYST acetyltransferase mutants (e.g. *gcn5/hag1*).

As discussed above, we provide new genetic data showing that mutations in *HAG1/GCN5* and *HAG3* cause defects in wound-induced callus formation (Figures 6d-f, S11a-c). We described these findings in the “Results” section as follows:

“Among the 5 members constituting the GNAT-MYST family of histone acetyltransferases, we further found that those encoded by *HAG1/GCN5* and *HAG3* are required for wound-induced reprogramming, as mutations in these genes strongly interfere with callus formation at wound sites (Figures 6d-f). On the contrary, the impact of mutations in the genes coding for other members of the GNAT-MYST family histone acetyltransferases, *i.e.* *HAG2*, *HISTONE ACETYLTRANSFERASE OF THE MYST FAMILY 1 (HAM1)* and *HAM3*, is far milder, if not negligible (Figures S11b-d).”

We also discussed these data in the “Discussion” section:

“Our genetic analysis further showed that *HAG1/GCN5* and *HAG3* are the primary histone acetyltransferases within the GNAT-MYST family that are involved in wound-induced callus formation (Figure 6).”

9) Fig. 7c shows expression changes of selected genes after wounding and MB3 treatment, but the levels of H3 acetylation after the inhibition should be also shown.

We included new ChIP-seq data to show that MB3 does indeed inhibit histone acetylation (Figures 7a, 7d). We described these data with the following sentences in the “Results” section:

“To assess whether these phenotypic defects are indeed associated with alteration of histone acetylation, we performed ChIP-seq analysis on plants exposed to 100 μ M MB3 from 24 h before wounding and harvested at 0, 3 and 6 h after wounding. Since overall levels of H3K9/14ac are not reduced to the level detectable by western blot after the short-term exposure to MB3 (Figure S11e), we normalized its enrichment to histone H3 levels. As expected, however, MB3 reduces the H3K9/14ac relative enrichment level for a large proportion of genes that possess this mark in control plants (Figure 7a and Table S6). For comparison, we also examined the enrichment of H3K4me3 in MB3-treated plants and found that their levels are also affected for 40 to 50% of genes presumably as a consequence of dampening of H3 acetylation (Figure 7a and Table S6).”

“For four of these genes (*WIND1*, *RAP2.6*, *DREB2D*, *RAP2.6L*), alteration of their expression dynamics in MB3-treated plants correlates with a reduction of their H3K9/14ac levels (Figure 7d). We should note, however, that these reductions are relatively mild after the short-term MB3 treatment, which is consistent with our western blot data that did not detect a visible reduction in the overall H3K9/14ac levels. For *LBD16*, we did not observe a clear reduction of histone acetylation in MB3-treated plants (Figure 7d), implying that its wound-induced transcriptional activation is independent from GNAT-MYST-mediated histone acetylation.”

10) Some figures are not enough described in the text and/or figure legends and therefore difficult to interpret and follow the authors` conclusions: Fig.1c , how to interpret percent numbers and representation factor? , Fig. 2b , description of how the PCA was performed is not clear, and to me the biggest difference is between H3K27me3 and H3K36me3

For Figure 1c, we provided a more detailed explanation on what the percents and the representation factors represent in the “Figure legends” as follows:

“Percentages indicate the ratio of genes induced or repressed by wounding among all genes containing the given mark. Bold numbers indicate representation values, *i.e.* the ratio between the representation of marked genes among wound-induced or -repressed genes and their representation across all genes.”

For Figure S5a (former Figure 2b), we provided more detailed description of how the PCA was performed in the “Methods” section as follows:

“For PCA analysis of the marking level before wounding, the “pca” and “fviz_pca_biplot” function of the R packages “factoextra” and “FactoMineR” were used. The following variables, ranking of histone modification level for each mark, presence in cluster 1 to 8 as depicted in Figure 2, and ranking of expression level before wounding, were considered for each gene.”

As pointed out by the reviewer, the large difference between H3K27me3 and H3K36me3 confirms their expected anti-correlation, supporting the validity of our PCA analysis. We added the following sentence to point this out in the “Results” section:

“As shown in Figure S5a, within PC1 and PC2 that explain 53.6% and 17.2% of variance respectively, pre-wound H3K27me3 and H3K36me3 marks display the largest difference, thereby confirming their antagonistic behavior.”

Minor points:

- the kinetics of expression changes of some genes shown in fig. S1c is different in roots and hypocotyls and it should be mentioned in the text.

We added the following sentence in the “Results” section to describe this difference between roots and hypocotyls.

“In particular, key reprogramming regulators such as *ERF115*, *LBD16*, *PLT3*, *RAP2.6L* and *WIND1* are induced both in roots and hypocotyls, although the induction of *PLT3*, *RAP2.6L* and *WIND1* is more transient in roots (Figure S1c).”

- P.5, line 168 : the Authors comment on early induced genes (3h) but what about earlier time points ?

Genes induced within the first 3 h, those in clusters 1 to 6, are all enriched for genes acting in response to stress. We rephrased the corresponding sentence in the text as follows:

“GO analysis performed on the set of genes from each cluster revealed that genes induced relatively early, *i.e.* those up-regulated within the first 3 h, are enriched for genes acting in response to stress, while genes induced at later time points include those implicated in various metabolic processes (Figure S4).”

- P.7, please specify cluster numbers when Fig. 5Aa is mentioned

The relationship between H3K9/14ac, H3K4me3 and transcriptional activation is generally similar for all clusters, although the delayed increase in H3K4me3 is most apparent for genes in clusters 1 and 2. We thus revised the corresponding section in the “Results” section as follows:

“Importantly, we also observed a delayed increase in the level of H3K4me3 that often reaches its maximum after the transcriptional peak and this trend was most notable for the rapidly induced genes in clusters 1 and 2 (Figure 5a).”

- description of GO analysis on histone modifications (methods) is not clear

We provided more detailed description of our GO analysis in the “Methods” section as follows:

“In order to evaluate GO term enrichment among loci associated with different chromatin states before wounding, we compared the ratio of marked genes associated with selected GO terms to the prevalence of these GO terms in the whole genome via a hypergeometric test. Gene lists associated with each selected GO terms were retrieved from TAIR10.”

Reviewer #3 (Remarks to the Author):

In this study, Rymer and colleagues examine the changes in gene expression and chromatin state that occur after wounding in *A. thaliana* roots. Previous work had suggested that changes in chromatin state were important for regulating genes involved in the regenerative process, but it was unclear which marks played a role in this reprogramming or which factors were involved. The authors performed RNA-seq and ChIP-seq before wounding and at several timepoints post-wounding. They looked at several histone modifications that might play a role in regulation - H3K27me3, a repressive mark, and the activating marks H3K36me3, H3K4me3, H3K9/14ac and H3K27ac. The authors found that all of the acetylation marks tested, but none of the methylation marks, showed strong changes after wounding that correlated with gene expression changes. They also showed that inhibiting histone acetyltransferase activity with MB3 prevents callus formation and attenuates gene expression changes after wounding, consistent with histone acetylation playing an important role in regeneration after wounding. Surprisingly, H3K27me3 and H3K36me3 levels remained nearly constant at genes induced after wounding, suggesting these are not involved in wound response, while H3K4me3 patterns suggested that this mark may follow but not trigger the gene expression program associated with wounding.

Overall I felt that this was an interesting and well-presented paper with novel findings consistent with predictions drawn from previous work in epigenetic reprogramming and stress/hormone response in plants. The paper was well organized, easy to read, and generally very thorough and careful in its interpretation of the data. The results highlight several interesting new questions, and provide a starting point for future work to examine which histone acetyltransferases are involved in facilitating the wounding response, and how they recognize their target genes. I did have a few minor comments/suggestions, listed below.

1. I found Fig. 2B hard to interpret overall. I would suggest dropping 2B and expanding 2C to show the same panel three times: as-is, sorted by H3K27me3, and sorted by H3K36me3. This will show that it's only the levels of pre-wounding acetyl marks that correlate well with the timing of induction. Expression at 0h could also be added as a column in those panels.

As suggested by the reviewer, we provided three panels, sorted by either H3K27me3, H3K36me3 or H3K9/14ac (Figure 2b). We would like to keep the PCA plot (former Figure 2b) as it permits us to add statistical significance to these data. It is now displayed in Figure S5a. We discussed the new panels as follows in the “Results” section:

“We did not observe such a clear linearity when we ranked the induced genes according to other pre-wound marking levels (Figure 2b), suggesting that early transcriptional induction correlates best with pre-wound H3 acetylation.”

2. Fig. 4 shows only data for the 3665 wound-induced genes, which makes it hard to tell how unusual the distribution of the 8 categories shown is. It would be useful to add the same pie-chart for all (expressed) genes next to the one for the 3665 wound-induced genes in panel A. p-values could also be estimated, e.g. using a bootstrapping approach, for the number of genes in each of the 8 categories in panel A vs. what would be expected from a set of 3665 genes drawn at random. Those p-values could also be added to the text (line 226, etc.).

This is an interesting point but given that we do not discuss the difference between wound-induced and all (expressed) genes in this figure, we don't think adding additional pie-chart will be helpful. The main purpose of Figure 4 is to examine the correlation between combinatory histone modification and the timing of induction upon wounding (Figure 4b) and a pie chart in Figure 4a is shown only as a reference. As suggested by the reviewer, we added a statistical analysis to Figure 4, in the form of hypergeometric tests comparing the distribution of the epigenetic categories between all the wound-induced genes and each expression-timing cluster. We explained our approach as follows in the “Figure legends”:

“Spie chart representing the percentage of genes associated with pre-wound H3K9/K14ac, post-wound H3K9/14ac and/or post-wound H3K4me3 among genes within clusters 1 to 8. Genes are grouped based on their association with H3K9/14ac. The radii of the wedges correspond to the representation factor (hypergeometric test) of the epigenetic category in the cluster compared to its representation among all wound-induced genes. The black circle corresponds to a representation factor of 1 so that wedges inside the circle depict an underrepresented category and wedges that extend beyond the circle depict an overrepresented category. * $p < 0.05$, ** $p < 0.01$, *** $p < 0.001$ (hypergeometric test).”

3. Fig. 1D - The heatmap is currently colored by p-value, but the magnitude of the p-value is not a good measure of effect size. Instead, the heatmap could be colored based on representation factor, with a scale indicating what a value > 1 indicates vs. a value < 1 (I had to google the meaning of 'representation factor' in the context of hypergeometric tests, despite having used those tests often in the past, so others may not be familiar with the term). This will also help differentiate pairs of marks that co-occur from pairs that are anticorrelated. Significant vs. nonsignificant cells could be indicated using significance stars, for example.

Following the reviewer's suggestion, we colored Figure 1d based on the representation values and indicated significance using stars instead. We used the \log_2 of the representation values to produce an equal color scale for displaying both underrepresented and overrepresented marks. We explained the representation value as follows in the “Figure legends”.

“...representation values, *i.e.* the ratio between the representation of marked genes among wound-induced or -repressed genes and their representation across all genes.”

4. In Fig. 3D, y-axis refers to fold-change (FC). I assume it's indicating FC of induction after wounding relative to pre-wounding mark levels, but please indicate explicitly.

To indicate this explicitly, we rephrased the corresponding sentence in the “Figure legends” as follows:

“MA-plots with M (histone modification ratio) and A (histone modification average) displaying the fold changes in H3K27me3, H3K9/14ac, H3K27ac, H3K4me3 and H3K36me3 levels compared to pre-wound histone marking levels, for all genes at 1, 3, and 6 h after wounding. Color and size of dots represent fold change (log(FC)) and significance (FDR corrected p values (edgeR test)), respectively, of the transcriptional response compared to pre-wound transcript levels at the same time point.”

5. Another hypothesis for why H3K27me3 levels were unaffected at induced genes could be that marking by H3K27me3 helps rapidly turn some wound-induced genes 'off' after the initial burst, preventing erroneous transcription. An interesting future direction might be to look at induction of these genes in a PRC2 mutant, to see if wound-induced genes fail to get turned off in this background. Alternatively, contrasting the pattern of gene activation over the 6h time series for genes that gain acetylation and are marked by H3K27me3, vs those that gain acetylation but have no H3K27me3, might be interesting.

We agree that this is an interesting hypothesis and we would like to investigate this possibility in future studies. We added this idea in the “Discussion” section:

“Another plausible hypothesis is that H3K27me3 acts as a buffer and helps to rapidly dampen the expression of a set of wound-induced genes after initial transcriptional bursts. Examining the expression of wound-induced PRC2 target genes in *prc2* mutants should clarify whether PRC2 activity is required to turn off their gene expression.”

6. It is interesting that loss of H3K27me3 was not required for upregulation of marked genes or gain of acetylation. H3K27me3 and H3K27ac are mutually exclusive, since (as far as I know) the lysine can't be simultaneously methylated and acetylated - this would suggest that genes that gain H3K27ac should lose H3K27me3, but this appears to not be the case in these data. What specifically was the relationship between these two marks in the data? Was H3K27ac only gained over genes with no H3K27me3? It might be useful to add the H3K27ac tracks to Fig. 5D.

We have indeed observed co-marking of H3K27ac and H3K27me3 (Figure 1d). As these marks should be mutually exclusive, we hypothesized that this could reflect either that they are deposited on separate nucleosomes within the same locus in the same cell, or that they originate from different cells, given that we study a heterogenous cell population with our experimental setup. We plan to explore these possibilities in future studies, for instance, by using sequential ChIP-seq. We added the following sentence to discuss these issues in the “Discussion” section:

“Our data imply instead that genes co-marked with H3K27me3 and active marks are more likely to be wound-inducible. Whether these histone modifications actually co-occur within the same cell should be assessed by performing sequential ChIP analysis in the future⁷⁰, as these marks might be present in different cells among the heterogenous population we sampled.”

7. My impression is that histone acetylation is generally less "stable" than methylation - turnover rates for methyl marks are much slower than for acetyl marks (for example, see Mews et al. 2014 Mol. Cell Biol. paper on histone methylation/acetylation on emerging from quiescence). So, methylation is often a more long-term, stable mark, whereas acetylation can be added and then removed quickly. Then, it is perhaps not surprising that all of the responsive marks detected in this study, which used a relatively short time series (6h), were acetyl marks. Perhaps in general, acetylation might be more likely to be involved in environmental responses that require sudden, rapid activation of a gene expression program. Are there any other studies that have shown a role for acetylation in mediating stress or hormone responses in plants in contexts other than regeneration?

Histone acetylation has been linked to various stress or hormonal responses in plants (Asensi-Fabado *et al.*, 2016; Yamamura *et al.*, 2016; Park *et al.*, 2018; Zheng *et al.*, 2019 and Li *et al.*, 2019). It is thus very possible that acetylation-mediated transcriptional reprogramming is a general mechanism to respond to environmental or endogenous cues. We added the following sentences in the “Discussion” section:

“Previous studies have also shown that histone acetylation is more dynamic than methylation⁶² and it is involved in various stress and hormonal responses in plants^{63–67}. Our data therefore support the view that acetylation-mediated transcriptional modification is a general mechanism to swiftly respond to environmental or endogenous cues.”

REVIEWERS' COMMENTS:

Reviewer #1 (Remarks to the Author):

The Authors have improved the manuscript with the addition of the treatments with inhibitors.

Answers to other queries has been only cosmetic.

While the data reported is the result of valuable and well designed work (after this set of revision), the Authors observe the consequences of transcriptional changes at the level of chromatin marks. The Authors have reported the transcriptional changes of wounding before in bulk cell populations. What is reported was predictable from their previous studies and does not constitute a conceptual advance. A major advance would be for example to have the study performed at the single cell level to be able to find what is truly reprogrammed in the chromatin prior transcriptional changes.

Reviewer #2 (Remarks to the Author):

The Authors made substantial improvements of the manuscript and addressed all my concerns. I have only minor remark to the revised part concerning MB3 impact on acetylation levels (p.10, lines 383-389). Authors may consider if `reduction` is a proper term in this case. Blocking HAT activity rather impedes the increase in acetylation levels across studied genes.

Reviewer #3 (Remarks to the Author):

All my previous comments have been well addressed by the authors, and I have no additional comments or suggestions.

Response to reviewer's comments

Reviewer #2 (Remarks to the Author):

The Authors made substantial improvements of the manuscript and addressed all my concerns. I have only minor remark to the revised part concerning MB3 impact on acetylation levels (p.10, lines 383-389). Authors may consider if `reduction` is a proper term in this case. Blocking HAT activity rather impedes the increase in acetylation levels across studied genes.

Our response:

We rephrased the description of our ChIPseq results on MB3-treated plants as below.

“relative H3K9/14ac levels are reduced for a large proportion of genes in MB3-treated plants”